# Syngeneic natural killer cell therapy activates dendritic and T cells in metastatic lungs and effectively treats low-burden metastases

Shih-Wen Huang[1,2†], Yein-Gei Lai[1†], Hao-Ting Liao[1,3†], Chin-Ling Chang[1], Ruo-Yu Ma[1], Yung-Hsiang Chen[1], Yae-Huei Liou[1], Zhen-Qi Wu[1], Yu-Chen Wu[4], Ko-Jiunn Liu[4], Yen-Tsung Huang[5], Jen-Lung Yang[1], Ming-Shen Dai[6*], Nan-Shih Liao[1,2*]

[1]Institute of Molecular Biology, Academia Sinica, Taipei, Taiwan; [2]Molecular and Cell Biology, Taiwan International Graduate Program, Academia Sinica and Graduate Institute of Life Science, National Defense Medical Center, Taipei, Taiwan; [3]Department of Life Sciences, National Central University, Taoyuan, Taiwan; [4]National Institute of Cancer Research, National Health Research Institutes, Tainan, Taiwan; [5]Institute of Statistical Science, Academia Sinica, Taipei, Taiwan; [6]Department of Hematology-Oncology, Tri-Service General Hospital, Taipei, Taiwan

*For correspondence:
dms1201@gmail.com (M-SD);
nsliao@gate.sinica.edu.tw (N-SL)

†These authors contributed equally to this work

## eLife Assessment

In this **important** study the authors develop an elegant lung metastasis mouse model that closely mimics the events in human patients. They provide **convincing** evidence for the effectiveness of IL-15/12-conditioned NK cells in this design, which was also critical for the authors being able to conclusively reveal the T cell-dependency of NK-cell-mediated long-term control of experimental metastasis. Of note, an investigator-initiated clinical trial demonstrated that similar NK cell infusions in cancer patients after resections were safe and showed signs of efficacy, which is of promising clinical application value.

**Abstract** Natural killer (NK) cells can control metastasis through cytotoxicity and IFN-γ production independently of T cells in experimental metastasis mouse models. The inverse correlation between NK activity and metastasis incidence supports a critical role for NK cells in human metastatic surveillance. However, autologous NK cell therapy has shown limited benefit in treating patients with metastatic solid tumors. Using a spontaneous metastasis mouse model of MHC-I+ breast cancer, we found that transfer of IL-15/IL-12-conditioned syngeneic NK cells after primary tumor resection promoted long-term survival of mice with low metastatic burden and induced a tumor-specific protective T cell response that is essential for the therapeutic effect. Furthermore, NK cell transfer augments activation of conventional dendritic cells (cDCs), Foxp3-CD4+ T cells and stem cell-like CD8+ T cells in metastatic lungs, to which IFN-γ of the transferred NK cells contributes significantly. These results imply direct interactions between transferred NK cells and endogenous cDCs to enhance T cell activation. We conducted an investigator-initiated clinical trial of autologous NK cell therapy in six patients with advanced cancer and observed that the NK cell therapy was safe and showed signs of effectiveness. These findings indicate that autologous NK cell therapy is effective in treating established low burden metastases of MHC-I+ tumor cells by activating the cDC-T cell axis at metastatic sites.

## Introduction

Metastasis is the most common cause of cancer death, and effective therapies for established metastases are currently lacking (*Ganesh and Massagué, 2021*). Natural killer (NK) cells play a critical role in metastasis control, as the abundance of functional circulating or tumor-infiltrating NK cells is inversely correlated with metastasis incidence while being positively correlated with favorable prognoses for patients with or at risk of metastasis (*López-Soto et al., 2017*). NK cells are capable of recognizing tumor cells via various activating and inhibitory NK receptors (NKRs). Tumor cells often down-regulate the expression of ligands for inhibitory NKRs, yet express stress-induced ligands for activating NKRs. Upon encountering tumor cells, the reduced inhibitory NKR engagement and enhanced activating NKR engagement facilitate NK cell activation (*Jamieson et al., 2002*; *Kärre et al., 2005*). NK cells also express receptors for activating cytokines, such as IL-15, IL-12, IL-18, and IL-21, as well as receptors for suppressive factors such as TGF-β and prostaglandin E2 present in the tumor microenvironment (*Huntington et al., 2020*). The sum of signals transduced through the activating and inhibitory receptors determines NK cell activation. Activated NK cells kill engaged tumor cells via perforin, granzymes, and the death receptor ligands TNF-α, TRAIL, and FasL (*Smyth et al., 2005*). Activated NK cells also produce IFN-γ, which directly inhibits tumor cell growth and induces type 1 T helper cell ($T_H1$) and cytotoxic CD8$^+$ T cell responses that are critical for effective anti-tumor immunity (*Alspach et al., 2019*; *Martín Fontecha et al., 2004*). Moreover, intratumoral NK cells produce XCL1 and CCL5 to recruit type 1 conventional dendritic cells (cDC1) that cross-present cell-associated tumor antigen to CD8$^+$ T cells (*Böttcher et al., 2018*), and also produce Flt3L to increase the abundance of cDCs (*Barry et al., 2018*). However, the immunosuppressive microenvironment induced by tumor cells elicits NK cell and T cell dysfunction (*Melaiu et al., 2019*; *Thommen and Schumacher, 2018*).

Two anti-metastatic mechanisms of NK cells have been revealed to date, i.e., cytotoxicity and IFN-γ production. In the metastatic cascade, the epithelial-mesenchymal transition increases the susceptibility of tumor cells to NK cell cytotoxicity by modulating the expression of ligands for the activating NKRs NKG2D (*López-Soto et al., 2013*), E-cadherin and cell adhesion molecule 1 (*Chockley et al., 2018*), as well as MHC class I (MHC-I) molecules (*Chen et al., 2015*). The IFN-γ produced by NK cells was shown to be essential for natural resistance to lung metastasis in a B16 melanoma model (*Takeda et al., 2011*) and for the IL-12-mediated suppression of liver metastasis in a CT-26 model (*Uemura et al., 2010*) using *Rag* knockout mice that lacked T cells. Another study indicated that the anti-metastasis effect of IL-12 produced by Baft3-dependent DCs is mediated through IFN-γ production by NK cells without the involvement of T cells (*Mittal et al., 2017*). NK cells also sustain the dormancy of metastatic tumor cells seeded in distal organs through cytotoxicity and IFN-γ (*Correia et al., 2021*; *Malladi et al., 2016*). The anti-metastatic mechanisms described above do not involve T cells. Thus, whether T cell immunity plays a role in the anti-metastatic function of NK cells remains to be elucidated.

Despite the clear anti-tumor function of endogenous NK cells, adoptive transfer of ex vivo-expanded autologous NK cells has shown limited clinical benefit in treating patients with metastatic or locally advanced solid tumors, including melanoma, renal cell carcinoma, and digestive cancer (*Parkhurst et al., 2011*; *Sakamoto et al., 2015*). Both the immunosuppressive microenvironment induced by tumor cells (*Melaiu et al., 2019*) and heterogeneity of NK cell subpopulations and their respective functions (*Freud et al., 2017*) likely contribute to this therapeutic limitation. Given the vast array of approaches used for ex vivo expansion of NK cells (*Fang et al., 2017*), different methods may result in NK cells exerting different functions. We hypothesize that autologous NK cells with anti-tumor activities are effective in treating low burden metastases. In this study, we used syngeneic IL-15/IL-12-conditioned NK cells to treat established lung metastases after resection of the primary MHC-I$^+$ EO771 mammary tumor. We found that the NK cell therapy promoted long-term survival of mice with low metastatic burden in a CD8$^+$ T-cell-dependent manner, and induced tumor-specific protective immune memory. Furthermore, NK cell transfer augmented the activation of cDC1, cDC2 and T cells in metastatic lungs, which requires IFN-γ of the transferred NK cells. We also conducted an investigator-driven clinical trial of autologous NK cell therapy on six patients hosting four cancer types and found that the therapy is safe and shows signs of efficacy.

## Results

### Characterization of ex vivo-expanded murine NK cells

To generate NK cells with anti-tumor activity ex vivo, we cultured murine bone marrow (BM) cells with IL-15 and IL-12, since IL-15 is an NK cell growth and survival factor and both cytokines enhance IFN-γ production and the cytotoxicity of NK cells (*Kennedy et al., 2000*; *Lodolce et al., 1998*; *Marcenaro et al., 2005*). On average, 86% of the resulting cells were identified as TCRαβ⁻TCRγδ⁻CD19⁻NK1.1⁺ that express NKG2D, EOMES and T-bet (*Figure 1A*), representing characteristics of NK cells (*Gordon et al., 2012*; *Raulet, 2003*). Approximately 90% of the expanded NK cells were CD27⁺ with no-to-low levels of CD11b expression (*Figure 1A*). The proportion of NK cells expressing MHC-I-interacting NKRs, including activating Ly49D/H and inhibitory Ly49A/G2/I and NKG2A, was stable among independent cultures (*Figure 1B*). In terms of activation status, on average, 87% of the expanded NK cells expressed either intermediate or high levels of the activating receptor DNAM-1 and 82% of them displayed an activated B220⁺CD11c⁺ phenotype (*Figure 1C*). Consistently, they expressed the anti-tumor effectors IFN-γ, granzyme B, and TRAIL (*Figure 1D*). Co-culturing the NK cells with either EO771 breast adenocarcinoma cells that express MHC-I molecules and Rae-1 or B16F10 melanoma cells that express none of those molecules in vitro (*Figure 1—figure supplement 1*) resulted in dose-dependent tumor cell death (*Figure 1E*) and up-regulation of IFN-γ production by the NK cells (*Figure 1F*). Thus, the NK cells conditioned by means of IL-15 and IL-12 possess anti-tumor activities in vitro.

BM cells were cultured with IL-15 and IL-12 as described in Methods. (A) Analysis of NK cells (CD19⁻TCRαβ⁻TCRγδ⁻NK1.1⁺) among live expanded cells. Plots are representative of 20 independent cultures. (B)-(D) Expression of MHC-I-interacting activating and inhibitory NKRs, activation markers and anti-tumor effectors by the expanded NK cells. Representative flow histograms are presented for each molecule, and the compiled data from independent expansion cultures is indicated by the dot plots. (E) In vitro cytotoxicity of the expanded NK cells. CFSE-labeled EO771 or B16F10 cells (T, target) were co-cultured with expanded NK cells (E, effector) at the indicated E:T ratios. Tumor cell viability was analyzed by PI staining using FACS. One representative of three independent experiments is shown. Data represent mean ± SEM of triplicate wells. Statistical significance was determined by unpaired two-tailed Student's t-test: *p<0.05, **p<0.01, ***p<0.001 relative to tumor only. (F) Tumor cells stimulated IFN-γ production by the expanded NK cells. Sorted NK cells were cultured alone (NK cells) or together with indicated tumor cells at a ratio of 1:1. Levels of intracellular IFN-γ of NK cells were determined by staining with anti-IFN-γ or isotype control antibody, and analyzed by FACS. Representative histograms from one of two independent experiments and the relative difference in mean fluorescence intensity (ΔMFI) of the two experiments are shown. ΔMFI was calculated by subtracting the MFI value of isotype control staining from that of anti-IFN-γ antibody staining. Relative ΔMFI was calculated by dividing the ΔMFI of NK cells +tumor cells with the ΔMFI of NK cells.

Representative flow plots of MHC-I molecules and Rae-1 expression by EO771 and B16F10 cells from three independent experiments are shown.

### EO771 tumor-resected mice represent a model for established metastases to test NK cell therapy

Cancer recurrence occurs in a significant proportion of patients after primary tumor resection due to dissemination of tumor cells before and/or during surgery. To mimic that scenario, we established a syngeneic orthotopic breast cancer model using the MHC-I⁺ EO771 cell line that harbors numerous mutations (*Yang et al., 2017b*). At 21 days post-inoculation and consistent with the spontaneous lung metastatic property of the EO771 line (*Ewens et al., 2005*), we observed that ~85% of mice with a primary tumor weighing ≥95 mg had metastatic foci on the lung surface. The number and area of metastatic foci was positively correlated with tumor weight on day 21 (*Figure 2A*). Resection of the primary tumor and its sentinel lymph node (LN) on day 21 resulted in a long-term survival rate of 18%, with mortality of all remaining mice (*Figure 2B*). All dead mice exhibited metastases in the lung, of which 71% displayed tumor reappearance at the surgical site (data not shown). Therefore, this tumor resection model represents a model of established metastases caused by tumor cell dissemination before and arising from surgery. To apply adoptive NK cell therapy to this model, we examined if the transferred NK cells reach the lungs of tumor-resected mice. The expanded NK cells express the chemokine receptors CXCR3, CCR5, and CXCR6 (*Figure 2C*), and the lung of tumor-resected mice

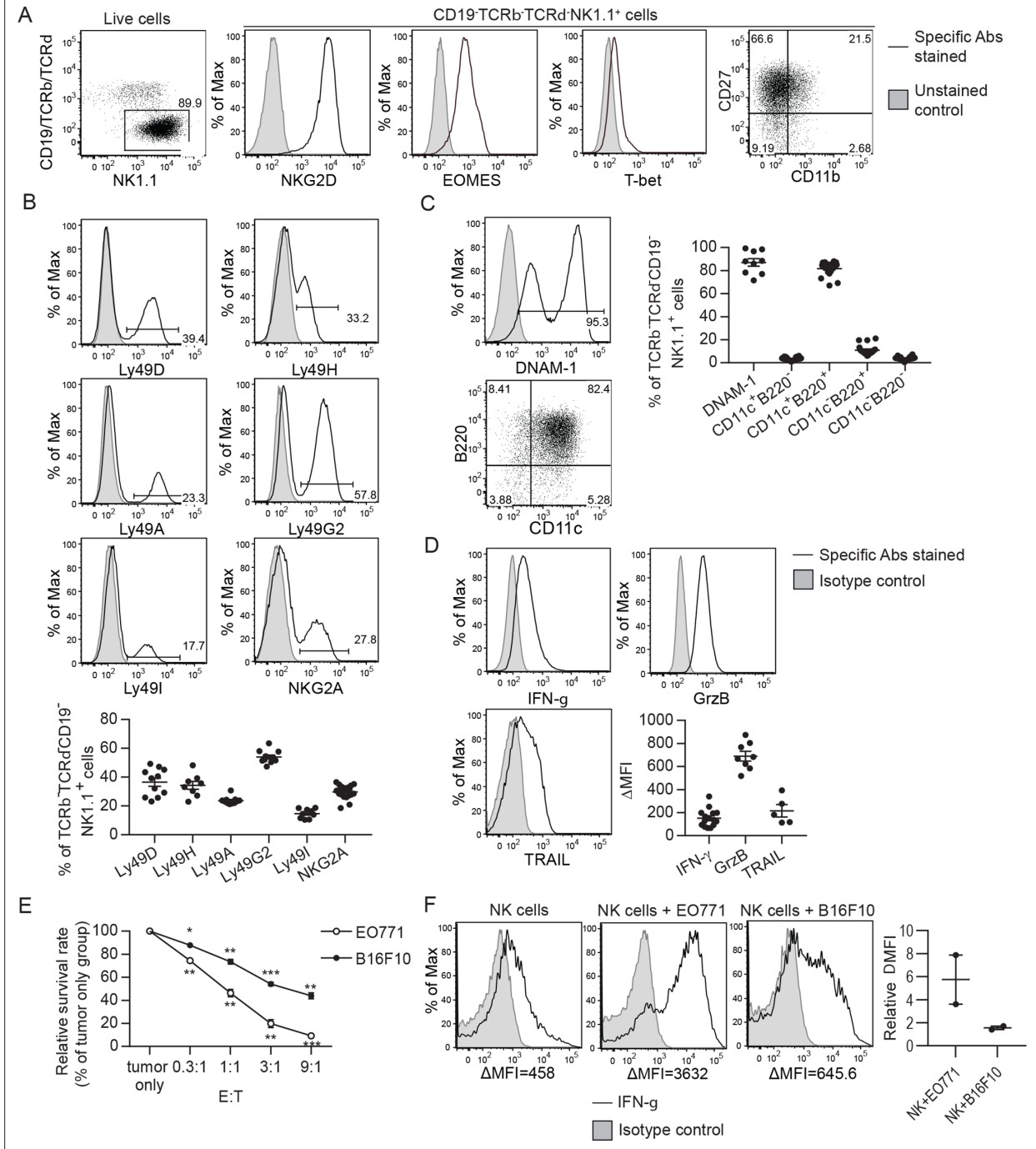

**Figure 1.** Characterization of ex vivo-expanded murine NK cells.

The online version of this article includes the following source data and figure supplement(s) for figure 1:

**Source data 1.** Expression of indicated NKRs, activation markers and anti-tumor effectors and in vitro cytotoxicity and IFN-g production in response to tumor cells by the expanded NK cells.

**Figure supplement 1.** Expression of MHC-I molecules and Rae-1 by EO771 and B16F10 cells in vitro (related to *Figure 1E*).

express mRNA of the corresponding ligands CXCL9/10/11, CCL3/4/5, and CXCL16 (*Figure 2D*). Using eGFP+ NK cells, we observed that numbers of transferred cells in the lung peaked at 4 hr and then continuously declined, remaining detectable for at least 2 days (*Figure 2E*). A similar pattern of transferred NK cells was observed in the spleen, albeit with lower cell counts (*Figure 2E*).

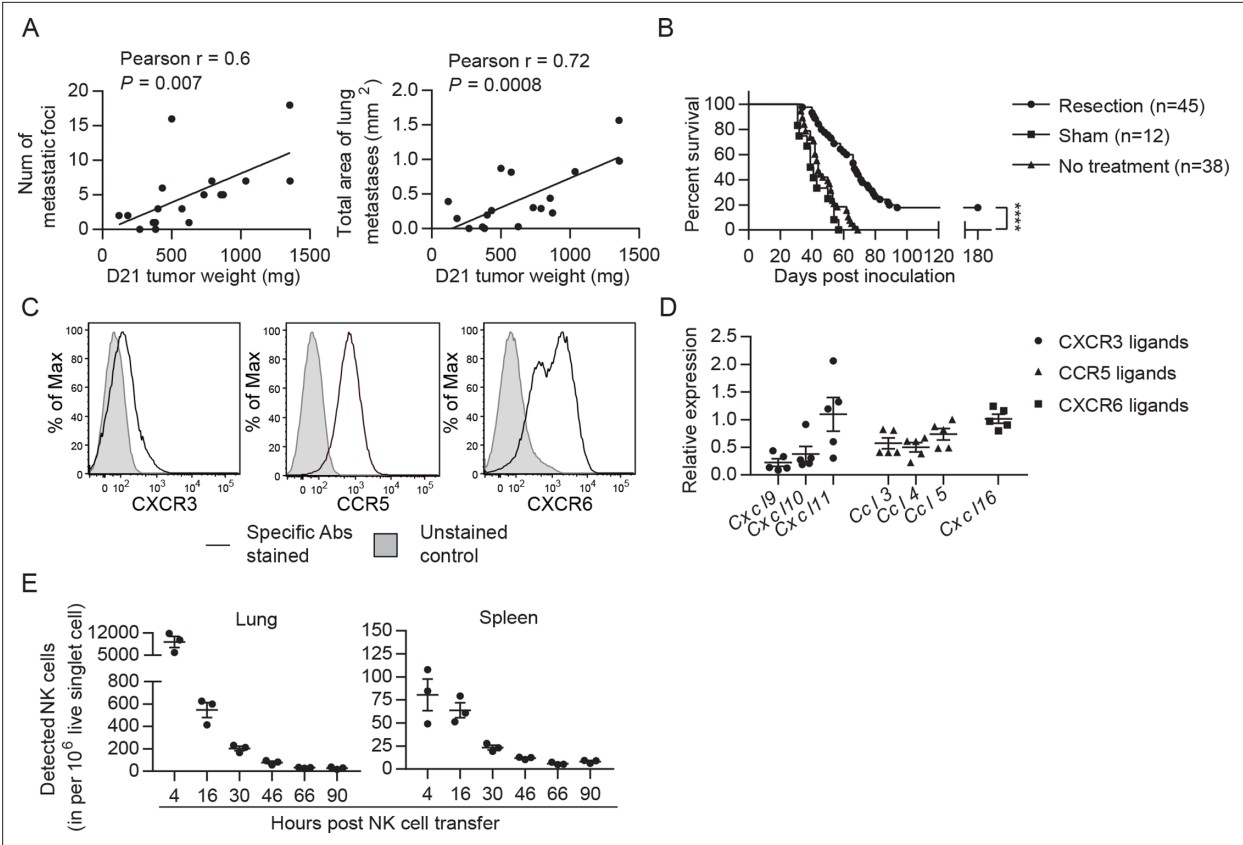

**Figure 2.** EO771 tumor resection and metastasis model for NK cell treatment. (**A**) Positive correlation between primary tumor weight and lung metastasis. Pearson correlations between day 21 tumor weights and the number and total area of metastatic foci on the lung surface are shown. Each dot represents a mouse. (**B**) Survival of mice who received day 21 tumor resection, sham surgery, or no treatment. Mouse survival was analyzed by means of the Kaplan-Meier estimator and a Log-Rank test: ****p<0.0001. (**C**) Expanded NK cells express the chemokine receptors CXCR3, CCR5, and CXCR6. Representative flow plots of 10 independent cultures are shown. (**D**) Expression of CXCR3, CCR5, and CXCR6 ligands in the lung. Specific gene expression was determined by quantitative real-time PCR using lung RNA of day 21 tumor-resected mice collected on day 24 post tumor inoculation and normalized to the expression of cyclophilin a. Each symbol represents a mouse and means ± SEM are shown. (**E**) Expanded eGFP+ NK cells were transferred into day 21 EO771-resected mice on day 24, and quantified in the single cell suspensions of spleen and lung at the indicated times using FACS. Numbers of eGFP+ NK cells detected in each organ are shown. Each dot represents a mouse (means ± SEM).

The online version of this article includes the following source data for figure 2:

**Source data 1.** Correlation between tumor weight and lung metastasis, mice survival, expression of CXCR3, CCR5 and CXCR6 ligands, and the number of transferred EGFP+ NK cells.

## Syngeneic NK cell therapy is effective in treating mice with low metastatic burden in a CD8+-T-cell-dependent manner

We evaluated the effect of syngeneic NK cell therapy on tumor-resected mice (*Figure 3A* Schema), and found a negative correlation between survival time and day 21 tumor weight in the NK-cell-treated group but not in the control group (*Figure 3A*). Consequently, we divided the mice into two groups according to their day 21 tumor weight. NK-cell-treated mice displayed significantly enhanced overall survival (OS) and an improved long-term survival rate from 33% to 68% compared to control mice in the 95–400 mg tumor group, whereas the same therapy did not improve OS of the >400 mg tumor group (*Figure 3B*). These results, together with the result presented in *Figure 2A*, indicate that the NK cell therapy promotes long-term survival of tumor-resected mice carrying low metastatic burden. We further examined if the surviving mice after NK cell treatment had acquired tumor-specific protection by re-challenging them with either EO771 or B16F10 tumor cells (*Figure 3C* Schema). We found that 83% of these mice were protected from EO771 re-challenge and survived beyond 180 days, whereas all mice re-challenged with B16F10 died (*Figure 3C*). Age-matched naïve mice inoculated for the

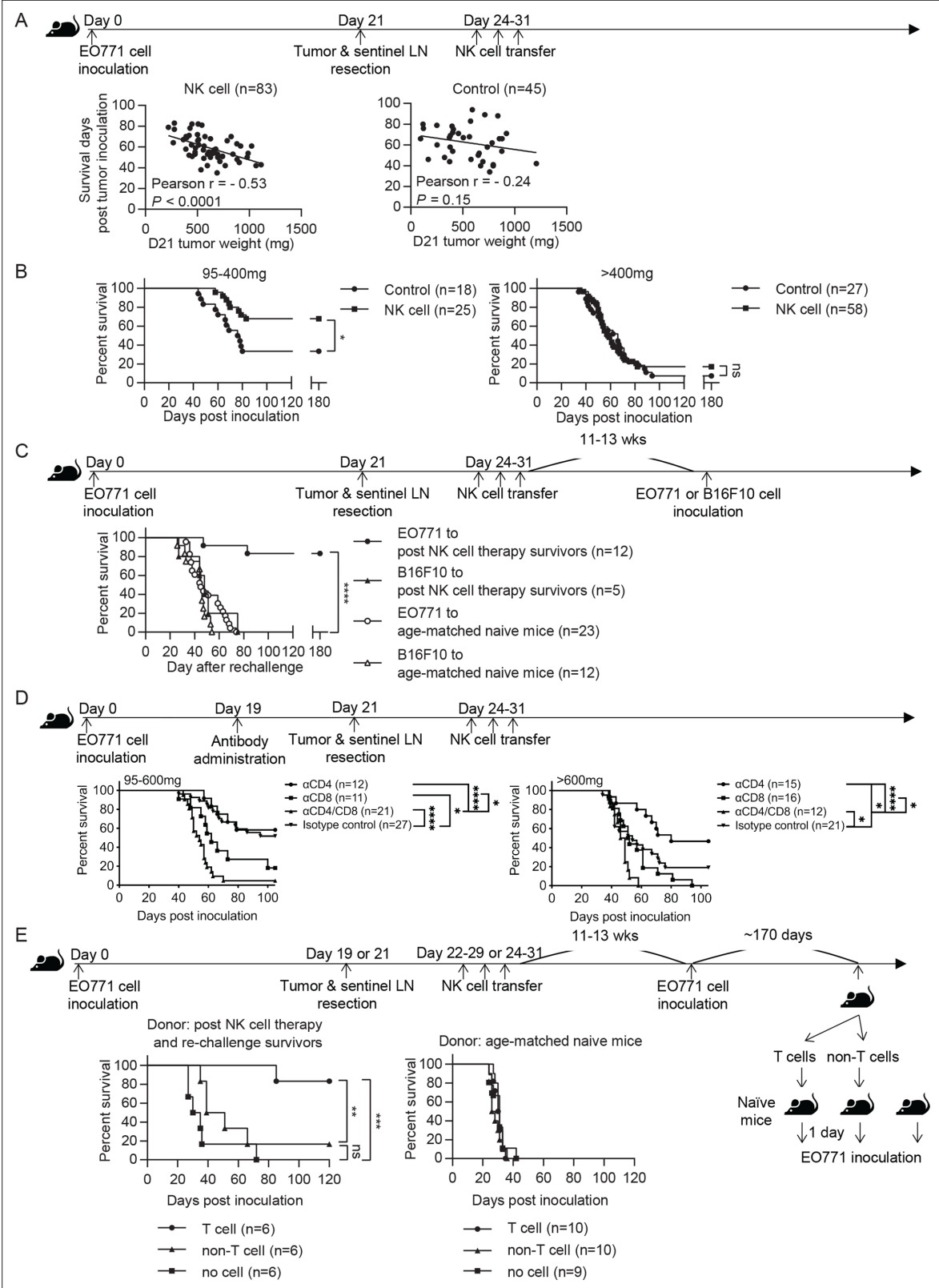

**Figure 3.** Syngeneic NK cell therapy is effective in treating mice with low metastatic burden in a CD8[+]-T-cell-dependent manner. (**A**) Day 21 tumor weight is inversely correlated with survival duration of NK-cell-treated tumor-resected mice. After tumor and sentinel LN resection on day 21, mice received either sorted NK cells (NK cell) or PBS (Control) at the indicated time-points. The correlation between day 21 tumor weight and survival time of mice was evaluated by Pearson correlation. Each dot represents a mouse. The results have been compiled from five to eight independent experiments.

*Figure 3 continued on next page*

*Figure 3 continued*

The Control group is the same dataset as presented for the Resection group in *Figure 2B*. (**B**) NK cell therapy promotes long-term survival of mice who had light tumors on day 21. The mice described in (**A**) were separated into 95–400 mg and >400 mg day 21 tumor weight groups for survival analysis by means of the Kaplan-Meier estimator and a Log-Rank test: *p<0.05. (**C**) NK cell therapy induces tumor-specific protection. Surviving mice after NK cell therapy were re-challenged with EO771 (●, compiled from four independent experiments) or B16F10 cells (▲, compiled from two independent experiments) at 11–13 weeks post therapy, and then monitored for survival. Age-matched naïve mice were inoculated with EO771 (O, compiled from four independent experiments) or B16F10 cells (△, compiled from three independent experiments) as controls. Mouse survival, using 2000 mm³ tumor volume as a cut-off, was analyzed by means of the Kaplan-Meier estimator and a Log-Rank test: ****p<0.001. (**D**) Depletion of CD4$^+$ or/and CD8$^+$ cells alter the effects of NK cell therapy. Mice were treated with indicated antibodies at day 19 post tumor inoculation, and then received resection followed by NK cell therapy. Mice were grouped according to indicated primary tumor weight for survival analysis by means of the Kaplan-Meier estimator and a Log-Rank test: ns, not significant; *p<0.05; ***p<0.001; ****p<0.0001. The data are compiled from two to three experiments. (**E**) T cells isolated from surviving mice post NK cell therapy and tumor re-challenge confer anti-tumor activity. Post NK cell therapy and re-challenge survivors were collected from three independent experiments. T cells and non-T cells isolated from each survivor were transferred separately into naïve recipient mice in a 1-donor-to-1-recipient manner, whereas other naïve mice that received no cells served as a control (compiled from two independent experiments). T cells and non-T cells were also isolated from age-matched naïve donors and transferred separately into naïve recipient mice, whereas other naïve mice that received no cells served as a control (compiled from two independent experiments). The recipient mice were then inoculated with EO771 cells one day after cell transfer. Mouse survival, using 2000 mm³ tumor volume as a cut-off, was analyzed by means of the Kaplan-Meier estimator and a Log-rank test: ns, not significant; **p<0.01; ***p<0.001.

The online version of this article includes the following source data and figure supplement(s) for figure 3:

**Source data 1.** Correlation between tumor weight and survival and mice survival analyses.

**Figure supplement 1.** Effectiveness of in vivo depletion of CD4$^+$ and CD8$^+$ T cells by antibody (related to *Figure 3D*).

**Figure supplement 1—source data 1.** Reduction rate of CD4+ and CD8+ T cells after antibody treatment.

first time with either EO771 or B16F10 as a control all died (*Figure 3C*). Thus, surviving mice of post-resection NK cell therapy displayed tumor-specific protective immunity.

Since the tumor re-challenge experiment was performed at 11–13 weeks following NK cell treatment (*Figure 3C*), the observed long-lasting protection is likely mediated by memory T cells. Accordingly, we examined the role of T cells in NK cell therapy by depleting T cell subsets with anti-CD4 or/and anti-CD8 antibodies two days before primary tumor resection (*Figure 3D* Schema and *Figure 3—figure supplement 1*). In the 95–600 mg tumor group, depletion of CD8$^+$ cells alone or both CD4$^+$ and CD8$^+$ cells diminished the effect of NK cell therapy, whereas depletion of CD4$^+$ cells alone did not affect OS (*Figure 3D*). This result indicates that CD8$^+$ T cells are essential for the effect of NK cell therapy. In contrast, the >600 mg tumor group displayed a limited NK cell treatment effect as expected, but did exhibit an improvement in OS upon depleting CD4$^+$ cells alone (*Figure 3D*). As the proportion of lung Foxp3$^+$CD4$^+$ T cells in CD45$^+$ cells positively correlated with day 21 tumor weight (data not shown), depletion of Foxp3$^+$CD4$^+$ T cells by anti-CD4 antibody likely has a stronger effect in augmenting the immune response for the >600 mg tumor than the 95–600 mg tumor group. Moreover, both tumor groups showed diminished OS upon depletion of both CD4$^+$ and CD8$^+$ cells than was the case for depletion of CD8$^+$ cells alone, indicating a CD8$^+$ T cell-independent anti-tumor effect of CD4$^+$ T cells (*Figure 3D*).

Next, we examined whether the T cells in surviving mice after NK cell therapy and tumor re-challenge possess anti-tumor activity. T cells and non-T cells were isolated from individual NK cell-treated and EO771-rechallenged survivors or from age-matched donors, and then transferred into naïve recipients in a 1-donor-to-1-recipient manner. The recipient mice were inoculated with EO771 cells to evaluate the anti-tumor effects of the donor cells, whereas naïve mice received no donor cells and served as a control for tumor growth (*Figure 3E* Schema). For the mice receiving cells isolated from the survivors, 5 of 6 T-cell recipients and 1 of 6 non-T-cell recipients survived beyond 120 days post tumor inoculation (*Figure 3E*). In contrast, mice receiving either T or non-T cells isolated from age-matched naive donors all died. Mice that received no donor cells in both experiments also all died (*Figure 3E*). Together, these results indicate that the post-resection NK cell therapy induces a tumor-specific T cell response with memory that is essential for its treatment efficacy.

Mice were treated with CD4- or/and CD8α-specific antibodies at day 19 post tumor inoculation, and then underwent tumor resection at day 21, followed by NK cell therapy during days 24–31. Levels of circulating CD4$^+$TCRβ$^+$ cells in the blood were reduced by 96–100% during the period of NK cell transfer, and by 70% and 54%, respectively, at 1- and 2 weeks post NK cell transfer. The level of

blood-circulating CD8$^+$TCRβ$^+$ cells was reduced by 99–100% during the period of NK cell transfer, and by 97% and 93%, respectively, at 1- and 2 weeks post NK cell transfer. Each time point compiles data from 3 to 10 mice from three independent experiments and is shown as mean ± SEM.

## Syngeneic NK cell transfer promotes cDC and T cell activation in metastatic lungs

Conventional DCs directly trigger an antigen-specific T cell response by providing ligands to antigen, costimulatory and cytokine receptors expressed on T cells (Cabeza-Cabrerizo, *Cabeza-Cabrerizo et al., 2021*). A previous in vivo study reported that A20 lymphoma expressing abnormally low levels of MHC I molecules (MHC-I) induces IL-12 production by splenic DCs (CD11c$^+$CD19$^-$F4/80$^-$) and CD8$^+$ T cell response (*Mocikat et al., 2003*). Therefore, we hypothesized that our NK cell therapy would modulate cDCs in metastatic lungs in favor of a T cell response. As our IL-15/IL-12-conditioned NK cells express high levels of mRNA encoding IFN-γ and IL-10 (data not shown), cytokines known to modulate cDC function, we examined whether one transfer of wild type (WT), *Il10*$^{-/-}$ or *Ifng*$^{-/-}$ NK cells affected cDCs and T cells in the metastatic lungs. Immune cells were analyzed ~18 hr after NK-cell transfer, by which time the level of transferred NK cells in the lung declined by >60% relative to that determined at 4 hr (*Figure 2E*). We found that lung cDC1s and cDC2s consist of cells expressing either high or low level of MHC-II (MHC-II$^{hi}$ or MHC-II$^{lo}$), and the MHC-II$^{hi}$ cDCs are composed of CCR7$^+$Lamp3$^+$ and CCR7$^-$Lamp3$^-$ subsets (*Figure 4A*). The CCR7$^+$Lamp3$^+$MHC-II$^{hi}$ subset expresses the highest level of APC function molecules, including MHC-I, MHC-II, CCR7, CD86, CD40, and PD-L1 (*Figure 4A*), resembling the recently identified mregDC (*Cheng et al., 2021*; *de Saint-Vis et al., 1998*; *Maier et al., 2020*). The CCR7$^-$Lamp3$^-$MHC-II$^{hi}$ subset expresses an intermediate level of MHC-II, CD86 and CD40 among the three subsets, and similar levels of MHC-I, CCR7 and PD-L1 to that of the MHC-II$^{lo}$ subset (*Figure 4A*). NK-cell transfer did not alter the proportion of total cDC1s or cDC2s in CD45$^+$ cells in the lung tissue (data not show). However, among cDC1s, transfer of WT NK cells significantly increased the proportion of the two MHC-II$^{hi}$ subsets, the levels of CD86 and CD40 expressed by all three subsets, and the levels of PD-L1, MHC-I and MHC-II expressed by CCR7$^-$Lamp3$^-$ MHC-II$^{hi}$ and MHC-II$^{lo}$ subsets (*Figure 4B*). IFN-γ, but not IL-10, of the transferred NK cells significantly contributes to these effects, except for the PD-L1 level in MHC-II$^{lo}$ subset (*Figure 4B*). Transfer of WT NK cells exerted many effects on cDC2s similar to those described above for cDC1s, albeit with several differences (*Figure 4B*). The differences include augmenting MHC-II level in the CCR7$^+$Lamp3$^+$MHC-II$^{hi}$ subset, reducing MHC-II level in the MHC-II$^{lo}$ subset, and having no effect on CD40 level in the CCR7$^+$Lamp3$^+$MHC-II$^{hi}$ subset (*Figure 4B*). IFN-γ of transferred NK cells significantly contributed to these effects of WT NK cell transfer on cDC2s, except for the increase of the CCR7$^+$Lamp3$^+$MHC-II$^{hi}$ subset, CD40 and MHC-II levels in the CCR7$^-$Lamp3$^-$MHC-II$^{hi}$ subset, and CD86 and MHC-II levels in the MHC-II$^{lo}$ subset (*Figure 4B*).

The experimental design is the same as depicted in the *Figure 3A* schema except that NK cells (WT, *Il10*$^{-/-}$ or *Ifng*$^{-/-}$) were transferred once on day 24 and lung immune cells were analyzed on day 25. (A) Gating strategies and characterization of cDC subsets in lung tissue. The dot plots show the gating of CCR7$^+$LAMP3$^+$MHC-II$^{hi}$, CCR7$^-$LAMP3$^-$MHC-II$^{hi}$ and MHC-II$^{lo}$ subsets of cDC1s and cDC2s. The histograms show the comparison of indicated molecules expressed by the three subsets of cDC1s and cDC2s. Negative controls are either fluorescence-minus-one (FMO) or single stain of another molecule. Data shown are from a representative PBS-treated mouse. (B) NK-cell transfer affects the activation of lung cDCs. The proportion of the three cDC subsets in cDC1s and cDC2s are shown. The expression of the indicated APC function molecules was analyzed as relative MFI, based on normalization against the mean MFI of the PBS group in each independent experiment. Data are compiled from two independent experiments and presented as mean ± SEM. Each symbol represents one mouse. Statistical significance was determined by unpaired two-tailed Student's t-test: *p≤0.05, **p≤0.01, ***p≤0.001.

Next, we examined T cells in the lung tissue. Transfer of WT NK cells increased the proportion of Foxp3$^-$CD4$^+$ and CD8$^+$ T cells without altering that of Foxp3$^+$CD4$^+$ T cells (*Figure 5A*, *Figure 5—figure supplement 1A*). The proportion of activated Foxp3$^-$CD4$^+$ T cells, including effector (CD62L$^-$CD44$^+$), PD-1$^+$ and Ki67$^+$ subsets, also increased (*Figure 5B*, *Figure 5—figure supplement 1A*). These effects were not observed for mice that received *Ifng*$^{-/-}$ NK cells (*Figure 5B*). CD8$^+$ T cells in a tumor microenvironment (TME) are known to display a complex composition comprising subsets of different

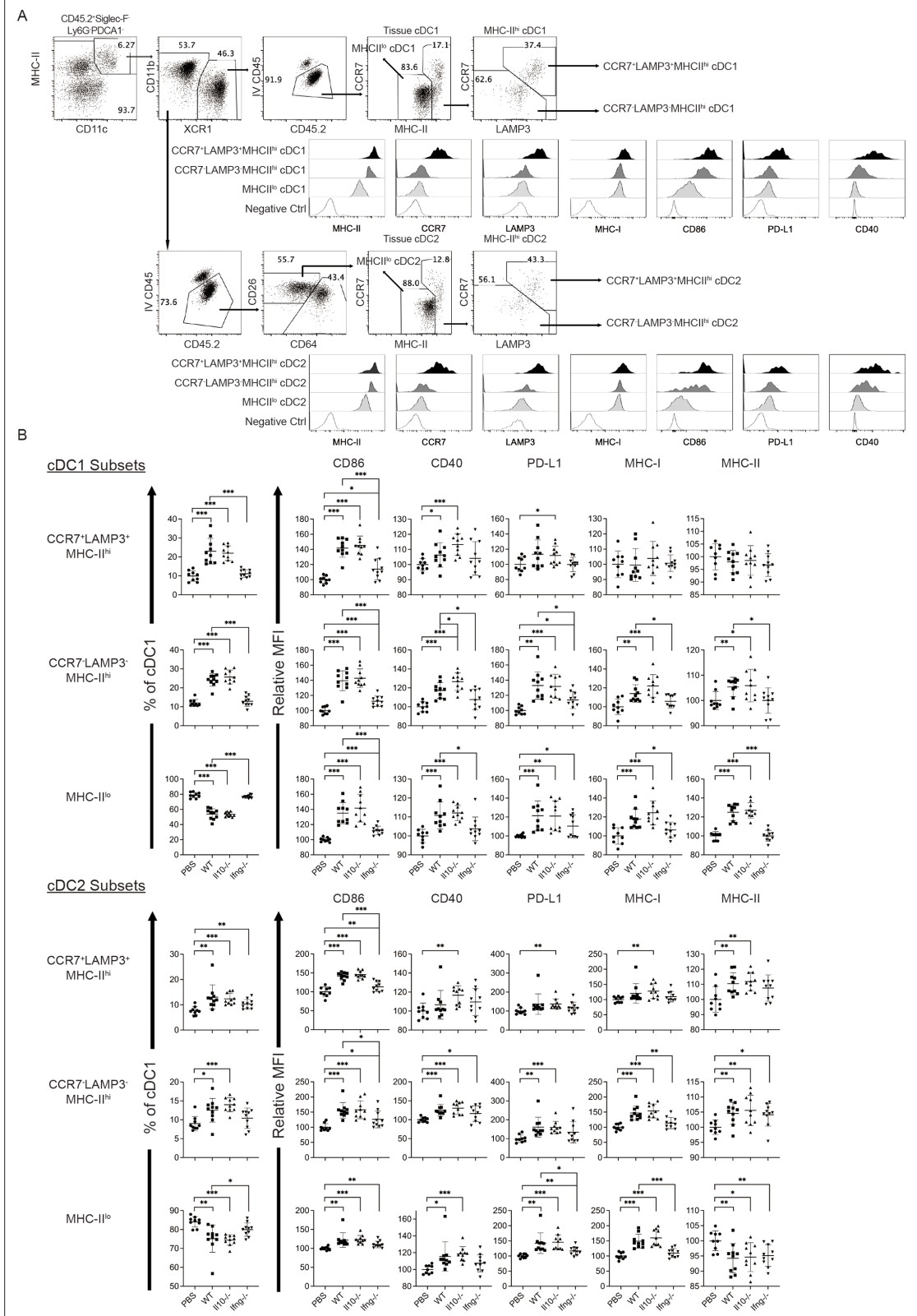

**Figure 4.** Syngeneic NK cell transfer modulates APC function of cDC in the metastatic lungs.

The online version of this article includes the following source data for figure 4:

**Source data 1.** Composition of cDC1 and cDC2 subsets and the expression of CD86, CD40, PD-L1, MHC-I and MHC-II by the cDC subsets.

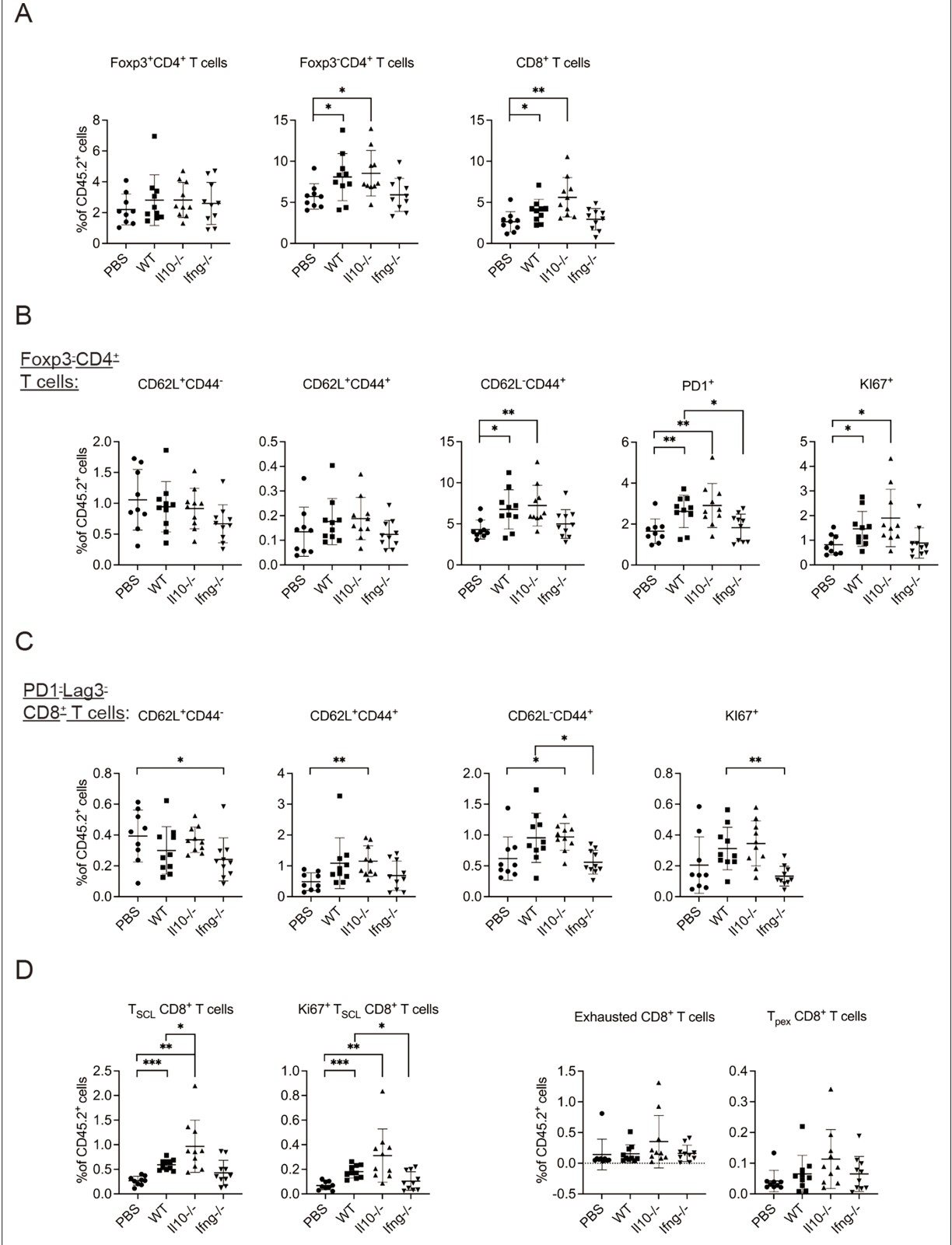

**Figure 5.** Syngeneic NK cell transfer promotes T cell activation in metastatic lungs.

The online version of this article includes the following source data and figure supplement(s) for figure 5:

**Source data 1.** The percentages of indicated T cell subpopulations in CD45+ cells.

**Figure supplement 1.** Analysis of T cells in lung tissue (related to *Figure 5*).

activation/differentiation statuses. We categorized lung tissue CD8[+] T cells according to PD-1 and Lag3 expression (*Figure 5—figure supplement 1A*), because PD-1 expression is induced by TCR stimulation (*Sharpe and Pauken, 2018*) and Lag3 is only expressed by exhausted or progenitor-exhausted (PEX) CD8[+] T cells (*Calagua et al., 2021*; *Yang et al., 2017a*). We reasoned that PD-1[-]Lag3[-] CD8[+] T cells (*Figure 5—figure supplement 1A*) are either naïve or at a very early stage of activation, and found that transfer of *Il10[-/-]*, but not WT, NK cells increased the proportion of CD62L[+]CD44[+] and CD62L[-]CD44[+] subsets, while the proportion of CD62L[-]CD44[+] and Ki67[+] subsets were lower in mice that received *Ifng[-/-]* NK cells compared to those in mice that received WT NK cells (*Figure 5C*). These results indicate that IL-10 of transferred NK cells inhibits activation of naive CD8[+] T cells, whereas IFN-γ of transferred NK cells promotes their effector differentiation and proliferation. We further examined stem cell-like (SCL) CD8[+] T cells that are Tim3[-] in PD-1[lo]Lag3[-] CD8[+] cells (*Figure 5—figure supplement 1A*; *Castiglioni et al., 2023*), and detected an increase of total and Ki67[+] SCL CD8[+] T cells in mice that received WT or *Il10[-/-]*, but not *Ifng[-/-]*, NK cells (*Figure 5D*). Transfer of *Il10[-/-]* NK cells resulted in a higher level of SCL CD8[+] T cells than when WT NK cells were transferred (*Figure 5D*). In contrast, NK cell transfer did not alter the proportion of exhausted or PEX CD8[+] T cells (*Figure 5D*, *Figure 5—figure supplement 1A*). The expression of PD-1, Lag3, Tim3, CD44, CD62L, Ly108 and GrzB by SCL, exhausted, and PEX CD8[+] T cells in our analysis is consistent with the known characteristics of these cell populations (*Figure 5—figure supplement 1B*; *Andreatta et al., 2021*; *Castiglioni et al., 2023*; *Martinez-Usatorre et al., 2020*). Thus, overall, transfer of the IL-15/IL-12-conditioned syngeneic NK cells augmented the activation of cDC1s and cDC2s, as well as the levels and activation of Foxp3[-]CD4[+] T cells, naïve CD8[+] T cells and SCL CD8[+] T cells in metastatic lungs. Moreover, IFN-γ of the transferred NK cells significantly contributes to these effects.

The experimental design is the same as *Figure 4*. The proportion of T cell subsets in CD45[+] cells in lung tissue were analyzed (*Figure 5—figure supplement 1*). (A) Foxp3[+]CD4[+], Foxp3[-]CD4[+] and CD8[+] T cells. (B) Foxp3[-]CD4[+] T cell subsets. (C) PD1[-]Lag3[-]CD8[+] subsets. (D) Total and KI67[+] SCL, exhausted, and PEX CD8[+] T cells. Data are compiled from two independent experiments and are presented as mean ± SEM. Each symbol represents one mouse. Statistical significance was determined by unpaired two-tailed Student's t-test: *p≤0.05, **p≤0.01, ***p≤0.001.

It is known that CCR7 mediates migration of cDC from tumor to its draining LN (dLN; *Riol-Blanco et al., 2005*), in which the migratory cDCs prime tumor-specific naive T cells (*Mempel et al., 2004*). As transfer of WT NK cells increased the levels of CCR7[+]LAPM3[+]MHC-II[hi] cDC1s and cDC2 in metastatic lungs (*Figure 4*), we hypothesized that NK cell transfer augments migration of lung cDCs to mLN and activation of T cells. We analyzed migratory and resident cDCs in mLN (*Figure 6—figure supplement 1*), and found that >90% of migratory cDC1s and >65% of migratory cDC2s express CCR7 and LAMP3 (*Figure 6A*), a phenotype similar to lung CCR7[+]LAMP3[+]MHC-II[hi] cDC1 and cDC2. On the other hand, the great majority of resident cDC1s and cDC2s in mLN are CCR7[-]LAMP3[-] (*Figure 6A*). Transfer of WT NK cells increased the proportion of migratory cDC1 in cDCs and the level of CD86 expression by both migratory cDC1s and cDC2s, which required IFN-γ of the transferred NK cells (*Figure 6B*). We also examined T cells in mLN and found that NK cell transfer affected neither the proportion nor the activation marked by CD44[+] or KI67[+] of Foxp3[-]CD4[+] and CD8[+] T cells (data not show). Collectively, NK cell transfer increased the level of migratory cDC1 and CD86 expression by migratory cDC1s and cDC2s, but did not affect T cell activation in mLN.

The experimental design is the same as *Figure 4*. (A) The expression of CCR7 and LAMP3 by migratory and resident cDC1s and cDC2s. (B) NK cell transfer affected migratory cDC subsets. The proportion of migratory cDC1s and cDC2s in cDC are shown. The expression of the indicated APC function molecules was analyzed as relative MFI, based on normalization against the mean MFI of the PBS group in each independent experiment. Data are compiled from two independent experiments and presented as mean ± SEM. Each symbol represents one mouse. Statistical significance was determined by unpaired two-tailed Student's t-test: *p≤0.05, **p≤0.01, ***p≤0.001.

Representative flow plots show the gating of migratory and resident cDCs in mLN.

## Clinical trial of autologous NK cell therapy on advanced cancer patients

To assess NK cell therapy in human, we expanded human NK cells from TCRβ[-]CD19[-] PBMCs. The expanded NK cells are identifiable as CD3[-]CD19[-]CD14[-]CD56[+]EOMES[+], with ~90% of the cells expressing HLA-DR (*Figure 7A*). Co-culture of the expanded HLA-DR[+] NK cells with K562, THP-1

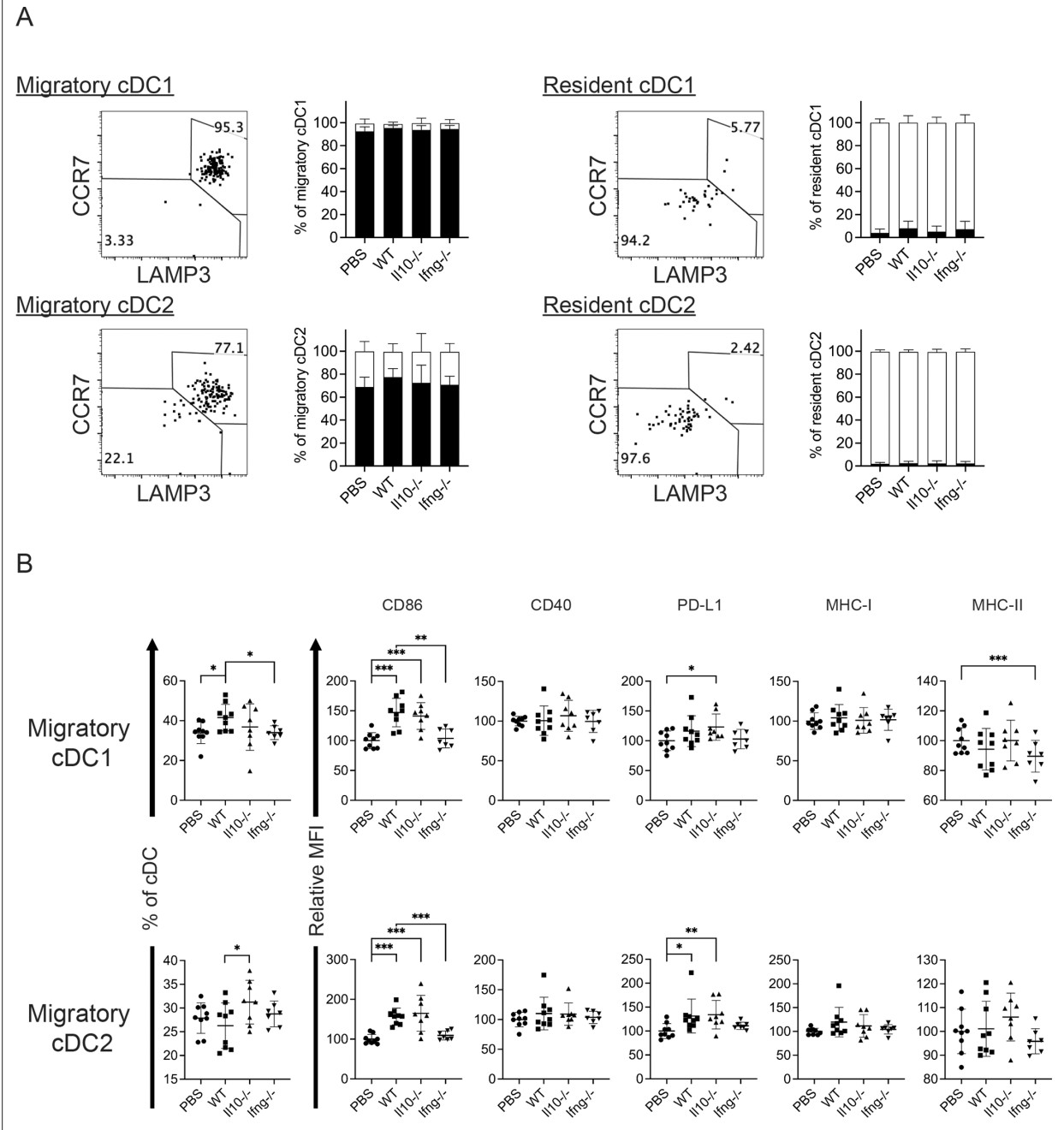

**Figure 6.** Syngeneic NK cell transfer affects migratory cDCs in mLN.

The online version of this article includes the following source data and figure supplement(s) for figure 6:

**Source data 1.** The expression of CCR7, LAMP3, CD86, CD40, PD-L1, MHC-I and MHC-II by indicated cDC subsets.

**Figure supplement 1.** Analysis of cDCs in mLN.

or U937 tumor cells resulted in dose-dependent tumor cell death (***Figure 7B***) and increased IFN-γ production by the NK cells (***Figure 7C***). We then conducted an investigator-initiated clinical trial of autologous NK cell therapy in six pre-treated advanced cancer patients enrolled between May 2016 and October 2019. Baseline characteristics of these patients are presented in ***Table 1***. The patients had an ECOG performance status of 0, a predicted survival of >3 months, and a relatively small target lesion size (6.4–22.7 mm), suggesting relatively low tumor burden. The patients in cohort 1 and cohort 2 received six bi-weekly infusions of 20×10⁶ or 30×10⁶ autologous HLA-DR⁺ NK cells, respectively.

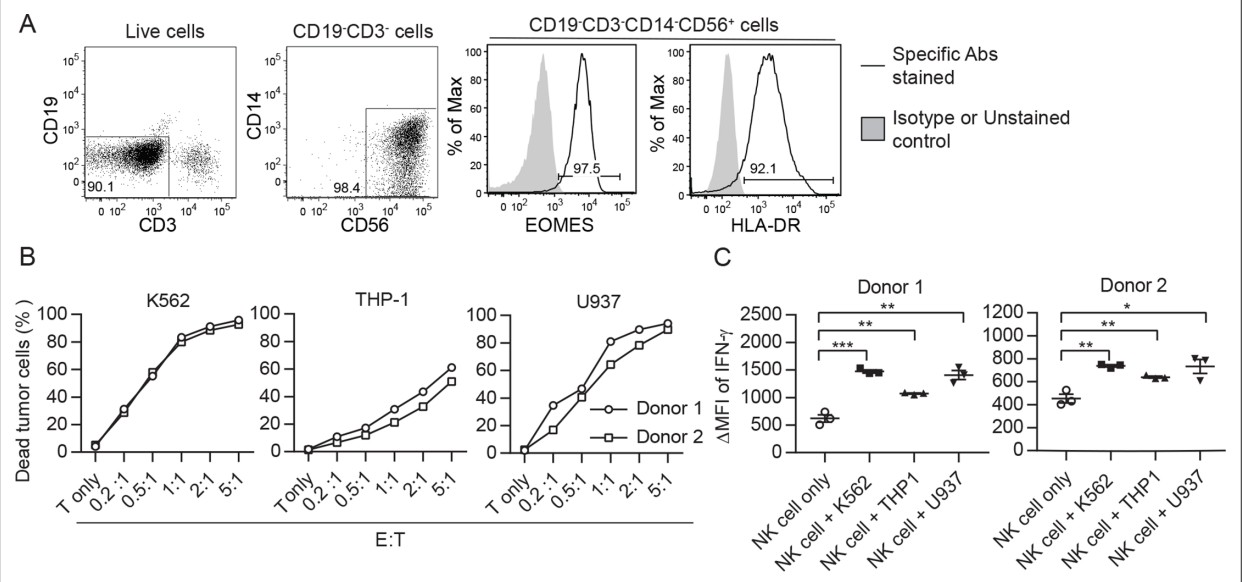

**Figure 7.** Ex vivo-expanded human NK cells exhibit anti-tumor activities in vitro. (**A**) Analysis of expanded human NK cells. Live cells among total cells obtained from human PBMC culture (as described in Methods) were analyzed for the expression of indicated molecules by FACS. Representative plots of 33 healthy donors are shown. (**B**) Expanded human NK cells kill tumor cells in vitro. Sorted HLA-DR⁺ NK cells from two healthy donors were co-cultured with CFSE-labeled tumor cells at the indicated E:T ratio. Percentage of dead tumor cells represents the mean ± SEM of PI⁺ cells among CFSE⁺ tumor cells from triplicate wells. (**C**) Expanded human NK cells up-regulate IFN-γ production in response to tumor cells in vitro. Sorted HLA-DR⁺ NK cells from two healthy donors were co-cultured with tumor cells at a 1:1 ratio for 5 hr. Levels of IFN-γ in NK cells were examined by intracellular staining and are calculated as the ΔMFI ± SEM between specific antibody-stained and isotype control antibody-stained samples. Statistical significance was determined by unpaired two-tailed Student's t-test: *p<0.05, **p<0.01, ***p<0.001 relative to NK cells only.

The online version of this article includes the following source data for figure 7:

**Source data 1.** The cytotoxicity and IFN-g production by expanded human NK cells in response to tumor cells in vitro.

The infused cells were total cells expanded from TCRβ⁻CD19⁻ PBMCs, of which median 92.4% (83.2–94.9%) were HLA-DR⁺ NK cells (*Figure 8A*, *Figure 8—figure supplement 1*). The composition of the infused cells differed slightly among patients, but was stable for each patient over the six infusions (*Figure 8A*). We examined the expression of NKRs, CD16 and CXCR3 on the HLA-DR⁺ NK cells and determined that it was stable for each patient over six batches of cell preparations, apart from some variation in NKp44 for patient 5 and CXCR3 for patient 4 (*Figure 8B*, *Figure 8—figure supplement 2*). Moreover, the HLA-DR⁺ NK cells expressed the anti-tumor effectors perforin and IFN-γ (*Figure 8C*, *Figure 8—figure supplement 3*). All patients completed the course of six infusions. The therapy proved safe and was well-tolerated by all patients, as no treatment-related adverse effect was observed throughout the entire infusion period up to 24 hr after the final infusion. We assessed the preliminary impact of this therapeutic approach based on clinical responses and OS, and present the results with a median follow-up of 60.0 months (2.73–60.0). The response and survival timings were counted from the first infusion of NK cells (*Figure 8D*). We adopted RECIST 1.1 to evaluate responses, except that the baseline target lesions for patients 3, 4, and 5 (*Table 1*) were 18%, 36% and 31% smaller than measurable lesions according to RECIST 1.1. All patients were in a stable disease (SD) state at month 1 (i.e. 1 week after the second infusion), and completed six infusions by month 2.3. Patients 1, 3 and 6 only received NK cell therapy, whereas patients 2, 4, and 5 received additional treatment(s) as indicated. Patient 1 exhibited progressive disease (PD) at month 2.7 and left the trial. Patient 2 displayed SD for 8.2 months, during which a partial response (PR) was detected at month 4.4. She then received maintenance metronomic Endoxan starting from month 8.4, and displayed SD or PR for a further 50 months. Patient 3 displayed SD for 52.7 months. Patient 4 exhibited an SD state for 13 months, received Tarceva during months 14.9–18.6, suffered PD at month 18.6 that prompted receipt of other treatments, and ultimately left the trial at month 31.5. Patient 5 presented SD for 9.9 months, and received metronomic Endoxan during months 7.7–13.5, 28.9–33.0, and 43.6–46.1. She underwent ultrasound imaging during months 17.6–54.2 and the results support a continuing

**Table 1.** Baseline characteristics of the patients enrolled in the phase I trial.

| Characteristic | Cohort 1 | | | Cohort 2 | | |
|---|---|---|---|---|---|---|
| Patient | 1 | 2 | 3 | 4 | 5 | 6 |
| Age | 63 | 42 | 59 | 66 | 54 | 65 |
| Gender | Male | Female | Female | Female | Female | Male |
| Primary cancer | Colon | Follicular lymphoma | Lung | Lung | Follicular lymphoma | Bladder |
| Metastasis | Adrenal Lung Pelvis | | Lung | Brain | | Bone |
| Previous therapies for cancer | | | | | | |
| Surgery | Yes | No | Yes | Yes | No | Yes |
| Radiotherapy | No | No | No | Yes | No | No |
| Lines of chemo- or targeted therapy | 1 | 3 | 0 | 3 | 2 | 4 |
| Immunotherapy | No | No | No | No | No | Keytruda[*] |
| ECOG performance status | 0 | 0 | 0 | 0 | 0 | 0 |
| Target lesion (mm) | Pelvic LN (16.3) | Left inguinal LN (18.4) | Mediastinal LN (12.3) | Lung left upper lobe (6.4) | Left axillary LN (10.4) | Pelvic LN (22.7) |

*Ended 7 months before NK cell therapy.

SD state. Patient 6 retained an SD state for 36.3 months, and subsequent ultrasound imaging in month 44.5 supports a continuing SD state. In summary, among the four patients with metastatic solid tumors, patients 1 and 4 exhibited PD at month 2.7 and 18.6, respectively, leaving the trial at months 2.7 and 31.5, respectively. In contrast, patients 3 and 6, who only received NK cell therapy, exhibited a SD state and both survived for 60 months. The two follicular lymphoma patients (patients 2 and 5) who started maintenance Endoxan therapy ~6 months after undergoing the NK-cell therapy have exhibited a SD state and both survived for 60 months. These preliminary results suggest some level of effectiveness for the autologous NK cell therapy, supporting the efficacy of our pre-clinical mouse model.

Flow plots representative of the six batches of cell preparation for each patient are shown.

Flow histograms representative of the six batches of cell preparation for each patient are shown. The solid line and the filled gray peak represent gated HLA-DR$^+$ NK cells stained with and without specific antibody, respectively.

Flow histograms representative of the six batches of cell preparation for each patient are shown. The solid line and the filled gray peak represent gated HLA-DR$^+$ NK cells stained intracellularly with specific antibody and isotype control antibody, respectively.

## Discussion

The limited benefit achieved with adoptive autologous NK cell therapy for metastatic solid tumors is paradoxical given the clear anti-tumor functions of endogenous NK cells. In this study, we found that transfer of syngeneic NK cells conditioned using IL-15 and IL-12 after resecting the primary tumor resulted in long-term survival of mice with low burden lung metastases. Our results indicate a requirement for CD8$^+$ T cells and the contribution of CD4$^+$ T cells to the effectiveness of this NK cell therapy (*Figure 3D*), which is distinct from the T-cell-independent anti-metastasis mechanisms of NK cells demonstrated previously (*Correia et al., 2021*; *Malladi et al., 2016*; *Mittal et al., 2017*; *Takeda et al., 2011*; *Uemura et al., 2010*). The difference in T cell dependency likely arises from differences in experimental design. These previous studies examined the effect of endogenous NK cells on experimental metastasis following intravenous or intrasplenic tumor cell injection by enumerating

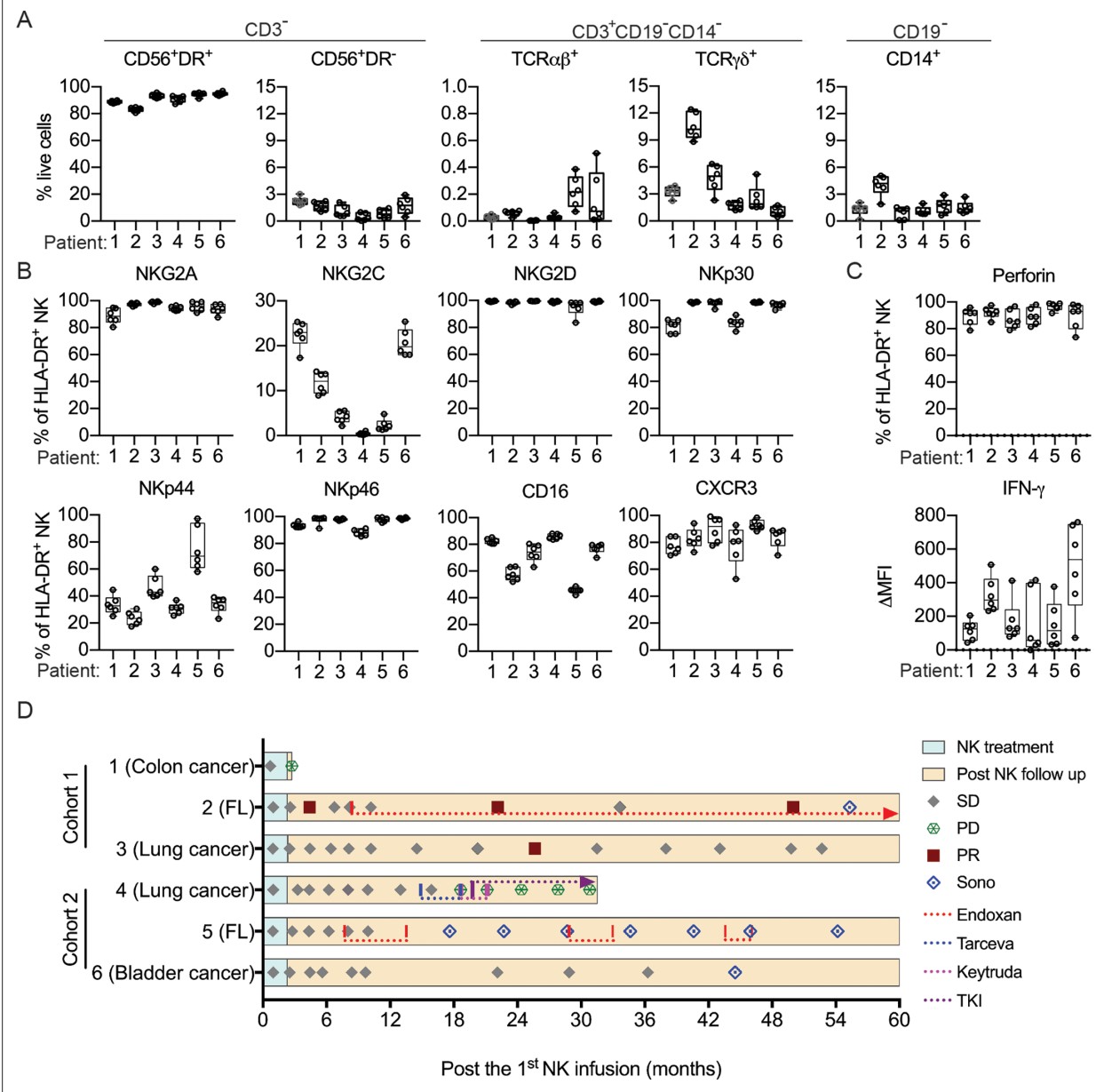

**Figure 8.** Clinical outcomes of autologous NK cell therapy. (**A**) Composition of the infused cells for each patient over six batches of cell preparation. Expression of (**B**) NKRs, CD16 and CXCR3 and (**C**) perforin and IFN-γ by the expanded HLA-DR⁺ NK cells over six batches of cell preparation. Each dot represents a batch of cell preparation. (**D**) Clinical responses and survival of the six patients. The blue and yellow blocks represent the durations of NK cell treatment and subsequent follow-up, respectively. The response of patients (SD, PD, or PR) was evaluated by CT imaging according to RECIST 1.1. Sono symbol marks follow-up using ultrasound imaging. Colored dotted lines indicate the timeframes for additional medications.

The online version of this article includes the following source data and figure supplement(s) for figure 8:

**Figure supplement 1.** Expression of CD56 and HLA-DR by CD3⁻CD19⁻CD14⁻ cells after ex vivo expansion (related to *Figure 8A*).

**Source data 1.** Composition and expression of NKRs, CD16, CXCR3, perforin and IFN-g by the expanded human NK cells.

**Figure supplement 2.** Expression of NKRs, CD16 and CXCR3 by the expanded HLA-DR⁺ NK cells (related to *Figure 8B*).

**Figure supplement 3.** Expression of IFN-γ and perforin in the expanded HLA-DR⁺ NK cells (related to *Figure 8C*).

metastases of affected organs 14 days later. Given that the effect of NK cells on experimental metastasis is mediated by clearing intraluminal tumor cells in microvessels before they extravasate into the target tissue (*Spiegel et al., 2016*), these studies assessed T cell-independent control of metastasis by NK cells. In contrast, our study examined the effect of adoptively transferred ex vivo-stimulated

NK cells on spontaneous metastasis by measuring long-term survival, in which lung metastases were established before primary tumor resection and NK cell administration (*Figure 2A*). The T cell dependency of the NK cell therapy in our model indicates that innate mechanisms are not sufficient to support the long-term survival of mice with EO771 metastases, which is consistent with the finding from experimental metastasis models that NK cells alone cannot reduce metastases after tumor cells have seeded the target tissue (*Spiegel et al., 2016*). Importantly, the timeframe we adopted for measuring long-term survival covers the period for the innate effect of the transferred NK cells and the effect of a full T cell response with memory. The CD8$^+$ T cell dependency implies that the transferred NK cells act as an upstream controller for an effective anti-tumor T cell response, which is in line with a previous study showing that early intratumoral accumulation of IFN-γ-producing endogenous NK cells is required for CD8$^+$ T cell-mediated eradication of *Cox*-deficient melanoma in immunocompetent wild-type mice (*Bonavita et al., 2020*).

In studying the mechanism underlying T cell-dependency for the effectiveness of NK cell therapy, our results from a mouse model reveal that transfer of WT NK cells augments activation of cDC1s and cDC2s in the metastatic lung (*Figure 4*). cDC1s and cDC2s present cell-associated and soluble antigens, respectively, to CD4$^+$ and CD8$^+$ T cells (*Hildner et al., 2008*). The increase of MHC-II$^{hi}$ cDC subsets and elevated expression of MHC-I, MHC-II and CD86 by lung cDC1s and cDC2s presumably promotes T cell activation, which is consistent with the increase of activated lung Foxp3$^-$CD4$^+$ and CD8$^+$ T cells (*Figure 5*). Moreover, the elevated CD40 expression by cDC1 and CD4$^+$ T cell activation implies cDC1-relayed CD4$^+$ T cell help for CD8$^+$ T cell response in the lung (*Ferris et al., 2020*). We also detected an increase of SCL CD8$^+$ T cells in metastatic lungs upon NK cell transfer (*Figure 5D*). Intratumoral tumor-specific SCL CD8$^+$ T cells are known to mediate the responses to PD-1/PD-L1 blockade and adoptive T cell immunotherapies (*Krishna et al., 2020*; *Siddiqui et al., 2019*) and are supplied from a reservoir of precursors in the tumor dLN (*Connolly et al., 2021*; *Schenkel et al., 2021*). As cDC1s in tumor dLN maintain the SCL CD8$^+$ T cell precursor reservoir (*Schenkel et al., 2021*), the increase of migratory cDC1 in mLN upon NK cell transfer (*Figure 6B*) likely contribute to the increase of lung SCL CD8$^+$ T cells. Moreover, a recent study reported that the SCL CD8$^+$ T cell precursors are activated by tumor antigens in the tumor dLN and acquire a stem-like state without an effector phenotype (*Prokhnevska et al., 2023*). After migrating to a tumor, they differentiate from SCL to an effector state, which requires costimulation by CD80 and CD86 of APC in the TME (*Prokhnevska et al., 2023*). Our findings of augmented CD86 expression by cDCs and increased proliferating SCL CD8$^+$ T cells in response to NK-cell transfer in metastatic lung are in line with the two-step activation of intratumoral SCL CD8$^+$ T cells. Together, these results demonstrate a clear association between the activation of lung cDCs and T cells in response to NK cell transfer. Furthermore, IFN-γ of transferred NK cells is essential for the augmented activation of both cDCs and T cells, strengthening evidence for the scenario that NK cells activate cDCs and then cDCs activate T cells. The requirement for IFN-γ of transferred NK cells also suggests a direct interaction between transferred NK cells and lung cDCs, which is supported by the increase of CCR7$^+$LAMP3$^+$ MHC-II$^{hi}$ lung cDCs according to a recent study reporting colocalization of LAMP3$^+$ DCs with NK cells in four types of human solid tumor (*Tang et al., 2023*). Moreover, this result indicates that NK-cell-derived IFN-γ drives the generation of mregDCs. However, although *Ifng$^{-/-}$* NK cells induced less cDC activation compared to WT NK cells, the levels of CD86 on cDCs of mice that received *Ifng$^{-/-}$* NK cells were higher than those of mice not subjected to NK cell transfer (*Figure 4B*). This outcome indicates the presence of IFN-γ-independent or/and compensatory mechanism(s) for cDC activation by the transferred NK cells, which is in line with our preliminary result that *Ifng$^{-/-}$* NK cell therapy does not significantly diminish the pro-survival effect in comparison to WT NK cell therapy beyond 60 days after tumor cell inoculation (data not shown).

IFN-γ is known to upregulate the expression of PD-L1 (*Blank et al., 2004*). Here, we found that IFN-γ of transferred NK cells augments the level of PD-L1 on cDCs, and the proportion of mregDCs that express the highest level of PD-L1 among lung cDC subsets (*Figure 4*). PD-L1 is a ligand for the inhibitory PD-1 receptor that induces Treg cell expansion and inhibits CTL cytotoxicity (*Arasanz et al., 2017*; *Dong et al., 2020*). A previous study found that PD-L1 on DCs inhibits T cell activation (*Peng et al., 2020*). Blockade of the PD-L1/PD-1 axis not only promotes activation of tumor-infiltrating CD8$^+$ T cells that leads to tumor control, but also improves the number and functionality of SCL CD8$^+$ T cells within tumors (*Castiglioni et al., 2023*; *Peng et al., 2020*; *Siddiqui et al., 2019*). Additionally, a recent study demonstrated that transfer of NK cells isolated from the spleens of 7-week-old mice,

combined with anti-PD-L1 treatment, restored the anti-tumor effect of anti-PD-L1 in aged mice of the MC38 tumor model (*Hou et al., 2024*). Collectively, we speculate that a combination of adoptive NK cell therapy plus PD-L1 blockade could enhance the effectiveness of cancer treatments.

In summary, our results indicate that syngeneic IL-15/IL-12-conditioned NK cell therapy promotes tumor-specific T cell responses, at least in part, through activation of cDC1s and cDC2s in metastatic lung. Our findings are consistent with the critical role of endogenous NK cells and CD8[+] T cells in anti-tumor immunity, and support that autologous NK cell therapy is effective in treating established low burden MHC-I[+] metastases by enhancing the tumor-specific CD8[+] T cell response.

## Materials and methods

### Mice and cell lines

Female C57BL/6JNarl mice (National Laboratory Animal Center of Taiwan, Taipei, Taiwan), *eGFP* transgenic C57BL/6JNarl mice (a gift from Chin-Yen Tsai at Academia Sinica (AS)), *Ifng*[-/-] mice (strain # 002287, The Jackson Laboratory, Bar Harbor, ME, USA) and *Il10*[-/-] mice (Strain # 002251, The Jackson Laboratory) were housed in a specific pathogen-free animal facility at the Institute of Molecular Biology. Mice aged 8–12 weeks old were used for experiments, unless stated otherwise. All animal protocols were approved by the IACUC of AS (Protocol number 11-09-221). EO771 (CH3 Biosystems), K562 (a gift from Che-Kun James Shen at AS), THP-1 and U937 (gifts from Li-Chung Hsu at National Taiwan University) cells were cultured in RPMI-1640 (Gibco, Grand Island, NY, USA) containing 10% FBS (Hyclone, Marlborough, MA, USA), 20 mM HEPES (Sigma-Aldrich, St. Louis, MO, USA), 100 U/ml penicillin and 100 μg/ml streptomycin (Gibco). B16F10 melanoma (a gift from Steve Roffler at AS) was cultured in DMEM (Gibco) containing 10% FBS and 100 U/ml penicillin and 100 μg/ml streptomycin.

### Flow cytometry

To stain cell surface molecules, cells were incubated with fluorophore-conjugated antibodies for 15 min at 4 °C and washed twice with staining buffer (PBS containing 2% FBS, 5 mM EDTA and 0.01% NaN$_3$). For staining of intracellular molecules, cells were fixed for 30 min at 4 °C after surface staining with 4% paraformaldehyde, or with a Foxp3/Transcription Factor Fixation/Permeabilization Concentrate and Diluent kit (Thermo Fisher Scientific, Waltham, MA, USA) to stain transcription factors. The fixed cells were then washed with staining buffer, permeabilized with 0.1% saponin, and stained with antibody for 30 min at 4 °C. Cells were analyzed by LSRII, FACSymphony A3 or FACSCalibur (BD Biosciences, Franklin Lakes, NJ, USA) and the data were analyzed using FlowJo (BD Biosciences). The antibodies used are detailed in the supplementary materials.

### Ex vivo expansion of murine and human NK cells

Murine BM cells were depleted of RBCs by means of ammonium-chloride-potassium (ACK) buffer (150 mM NH$_4$Cl, 10 mM NaHCO$_3$, 1 mM EDTA) and then cultured in RPMI-1640 containing 10% FBS (Corning, Corning, NY, USA), 20 mM HEPES (Sigma-Aldrich), 200 U/ml penicillin, 200 μg/ml strep-tomycin (Gibco), 50 μg/ml gentamycin (Sigma-Aldrich), 0.2 mg/ml L-glutamine (Sigma-Aldrich), and 50 μM 2-mercaptoethanol (Merck, Rahway, NJ, USA) in a 5% CO$_2$ incubator at 37 °C for 7 days with 30 ng/ml IL-15 (BioLegend, San Diego, CA, USA) and 10 ng/ml IL-12 (Peprotech, Cranbury, NJ, USA). The expanded murine NK cells (CD19[-]TCRβ[-]NK1.1[+]CD11c[+]B220[+]) were sorted using SORP (BD Biosciences). We used BM cells instead of splenocytes for NK cell culture because removal of T cells from BM cells before culturing is not necessary. Human NK cells were expanded from peripheral blood mononuclear cells (PBMCs) after depleting TCRβ[+] and CD19[+] cells with anti-TCRβ (WT31, BD Biosciences) and anti-CD19 (4G7, BD Biosciences) antibodies, and anti-mouse IgG1 microbeads (Miltenyi Biotec, Bergisch Gladbach, Germany) using an LD column with a QuadroMACS Separator (Miltenyi Biotec) under good tissue practice (GTP) conditions. The resulting TCRβ[-]CD19[-] PBMCs were cultured in RPMI-1640 containing autologous plasma, 20 mM HEPES (Sigma-Aldrich), 2 mM L-glutamine (Cellgro, Lincoln, NE, USA), and 50 μg/ml gentamycin (Winston Medical Supply Co. Ltd.) in a 5% CO$_2$ incubator at 37 °C for 7 days with 25 ng/ml IL-15 (CellGenix, Freiburg, Germany) and 10 ng/ml IL-12 (BioLegend; *Lee and Liao, 2015*).

### In vitro anti-tumor activity of expanded NK cells

Tumor cells were labeled using a Vybrant CFDA SE (CFSE) cell tracer kit (Invitrogen, Waltham, MA, USA). For cytotoxicity assay, sorted NK cells were co-cultured with 10[5] labeled tumor cells at the

indicated ratio and then incubated for 5 hr in a 5% $CO_2$ incubator at 37 °C. The cells were then stained with propidium iodide (PI) and the CFSE$^+$ tumor cells were analyzed for PI$^-$ cells using a LSRII system. For IFN-γ production, sorted NK cells were co-cultured with CFSE-labeled tumor cells at a 1-to-1 ratio, with 10 µg/ml brefeldin A being present for the last 4 hr of the co-culture. Cells were then stained intracellularly with anti-IFN-γ or isotype control antibody to detect IFN-γ production by CFSE$^-$ NK cells using a LSRII system.

## Trafficking of ex vivo-expanded murine NK cells

NK cells expanded from the BM cells of *eGFP* transgenic mice were sorted and intravenously transferred into C57BL/6JNarl mice (6 million cells/mouse) 3 days after resection of a day-21 tumor. Single-cell suspensions were prepared from the lung and spleen at the indicated hours after NK cell transfer and analyzed for eGFP$^+$ donor cells using a LSRII system.

## Gene expression

Lung RNA was extracted using Trizol (Invitrogen) and reverse-transcribed into cDNA using a Reverse Transcription Kit (SMOBIO, Paramount, California, USA). Relative gene expression was determined using a QuantStudio 12 K Flex Real-Time PCR system (Thermo Fisher Scientific) with SYBR Green and relative standard curves normalized against expression of cyclophilin a (*Cypa*). The sequences of primers used are detailed in the supplementary materials.

## EO771 resection and metastasis model and NK cell treatment

Each mouse was inoculated with 0.3 million EO771 cells into the right fourth mammary fat pad on day 0. Twenty-one days later, mice were anesthetized and the resulting tumor and sentinel LN (right inguinal LN) were resected. Sham control mice received the same surgical procedure without removing any tissue. Mice harboring a day-21 tumor weight of <95 mg were excluded since resection alone promoted long-term survival of mice with such tumors. Mice were grouped to have a similar mean and standard deviation of day-21 tumor weight. Each mouse then received 0.3 million expanded NK cells or PBS (Control) via the tail vein three times on days 24, 28, and 31, or once on day 24 as indicated. The resulting metastatic foci on lung surfaces were visualized by staining the dissected lung with India ink under a dissecting microscope. The area of metastatic foci was determined using ImageJ (NIH). For T cell depletion, mice were administered intraperitoneally with 200 µg of anti-CD4 (GK1.5, BioXcell, Lebanon, NH, USA), anti-CD8α (2.43, BioXcell), or both antibodies 19 days after EO771 inoculation, whereas control mice each received 400 µg of isotype control antibody (Rat IgG2b, κ; BioXcell). The mice underwent tumor resection 2 days later, before receiving NK cells as described in the previous section.

## Isolation of murine T cells and non-T cells

Single cell suspensions were prepared from the BM, spleen, and LNs of NK cell-treated and tumor-rechallenged long-term survivors and from the same tissues of age-matched naïve mice. After lysis of RBCs with ACK buffer, T cells and non-T cells were isolated using Pan T cell isolation Kit II (Miltenyi Biotec) via an autoMACS Pro Separator (Miltenyi Biotec) according to the manufacturer's instructions. The T cell preparations comprised 88–93% of CD3$^+$ cells, and the non-T cell preparations comprised 90–96% of CD3$^-$ cells.

## Preparation of lung cell suspension

Each mouse received an intravenous injection of 3 µg of FITC-anti-CD45 antibody three minutes before euthanasia to label blood CD45$^+$ cells. Lungs were harvested, minced and washed with PBS to remove excess anti-CD45 antibody. The tissue pieces were digested with 0.2 µg/ml of collagenase IV in HBSS (Sigma H1641) containing 2% FBS and 10 mM HEPES for 30 min at 37 °C with shaking at 200 rpm. RBCs in the lung cell suspension were lysed using ACK lysis buffer.

## Clinical trial design

We conducted an investigator-driven, open-label, 3+3 design trial to investigate the safety of the autologous NK cell therapy on metastatic/refractory stage IV cancer patients at a single center (ClinicalTrials.gov Identifier: NCT02661685). All six patients were ineligible for or refused further systemic

chemotherapy or immunotherapy, and had an Eastern Cooperative Oncology Group (ECOG) performance status of 0, a target lesion, and adequate organ function at the time of enrollment. The three patients in cohort 1 received six bi-weekly intravenous infusions of $20\times10^6$ expanded autologous HLA-DR$^+$ NK cells, whereas the three patients in cohort 2 were administered with $30\times10^6$ of the same cells. All patients underwent a computerized tomography (CT) scan between the second and the third NK cell infusions, and then five further CT scans at 8 week intervals starting one week after the sixth infusion. Thereafter, CT or sonography scans were performed as indicated. Safety was evaluated by assessing dose-limiting toxicity (DLT), defined as any treatment-related toxicity of grade 3 or above of general or immune disorders according to the National Cancer Institute-Common Terminology Criteria for Adverse Events (NCI-CTCAE) v4.03. Fever, chills, flu-like symptoms, or infusion-related reactions of grade 3 or above were considered DLT only if they remained at grade 3 or above for more than 3 days despite appropriate medication. Clinical responses to the therapy were evaluated by CT scans according to Response Evaluation Criteria in Solid Tumors version 1.1 (RECIST 1.1). Patient survival was followed every 8 weeks for the first year and every 12 weeks from the second to the fifth year.

## Trial oversight

The protocols for our clinical trial and follow-up were approved by the Institutional Review Boards (IRBs) at Tri-Service General Hospital and at AS, and were conducted according to the principles of the Declaration of Helsinki. Written informed consent was obtained from all patients.

## Statistical analysis

Results are presented as mean ± SEM. Statistical significance of in vitro cytotoxicity was examined by unpaired two-tailed Student's t-test. Correlation was determined by Pearson correlation coefficient. The Kaplan-Meier estimator was employed for survival analysis, and statistical significance was determined by a Log-Rank test. Tumor volume was calculated using the formula: length x width$^2$ x0.52. Mice with a tumor volume exceeding 2000 mm$^3$ were considered moribund. All statistical analyses were performed using GraphPad Prism 7 (GraphPad). ns, not significant; *$p<0.05$; **$p<0.01$; ***$p<0.001$; ****$p<0.0001$.

## Acknowledgements

We thank the FACS Core and Animal Facilities at the Institute of Molecular Biology, AS, for service and support. We thank Dr. Chee-Jen Chang at the Graduate Institute of Clinical Medical Science, Chang Gung University, for consultation on statistical design of the clinical trial. We thank Ms. Yi-Min Liu and colleagues at the Translation Resource Center for Genomic Medicine, Institute of Biomedical Science, AS, for assistance with designing the case report form, the Oracle clinical system and IRB affairs. We thank the Instrument Center at the National Defense Medical Center for flow cytometry services. We thank Dr. John O'Brien for manuscript editing.

## Additional information

### Competing interests

Nan-Shih Liao: The following patent information is related to the work presented in this study. US9157068 B2: Formulation for cultivating dendritic killer cells and method using the same; US9597356 B2: Method for treating cancers with dendritic killer cells and pharmaceutical composition comprising the same; Applicant: Academia Sinica; Inventor: NSL. The other authors declare that no competing interests exist.

### Funding

| Funder | Grant reference number | Author |
| --- | --- | --- |
| Ministry of Science and Technology, Taiwan | MOST 105-2325-B-016-002 | Nan-Shih Liao |

| Funder | Grant reference number | Author |
| --- | --- | --- |
| Academia Sinica | AS-TP-106-L08 | Nan-Shih Liao |
| Academia Sinica | Intramural funding | Nan-Shih Liao |

The funders had no role in study design, data collection and interpretation, or the decision to submit the work for publication.

### Author contributions
Shih-Wen Huang, Hao-Ting Liao, Formal analysis, Validation, Investigation, Methodology, Writing – original draft, Writing – review and editing; Yein-Gei Lai, Formal analysis, Validation, Investigation, Methodology, Writing – original draft, Project administration, Writing – review and editing; Chin-Ling Chang, Ruo-Yu Ma, Zhen-Qi Wu, Formal analysis, Validation, Investigation, Methodology; Yung-Hsiang Chen, Investigation; Yae-Huei Liou, Formal analysis, Investigation, Methodology; Yu-Chen Wu, Validation, Methodology; Ko-Jiunn Liu, Supervision, Validation, Methodology; Yen-Tsung Huang, Conceptualization, Formal analysis; Jen-Lung Yang, Formal analysis, Investigation; Ming-Shen Dai, Resources, Formal analysis, Supervision, Writing – original draft, Project administration; Nan-Shih Liao, Conceptualization, Resources, Supervision, Funding acquisition, Validation, Methodology, Writing – original draft, Project administration, Writing – review and editing

### Author ORCIDs
Yein-Gei Lai https://orcid.org/0000-0002-7342-4395
Hao-Ting Liao https://orcid.org/0009-0008-9048-7698
Nan-Shih Liao https://orcid.org/0000-0003-3707-4145

### Ethics
registration ClinicalTrials.gov Identifier: NCT02661685.
The protocols for our clinical trial and follow-up were approved by the Institutional Review Boards at Tri-Service General Hospital and at Academia Sinica, and were conducted according to the principles of the Declaration of Helsinki. Written informed consent was obtained from all patients.
All mouse protocols were approved by the IACUC of Academia Sinica. (Protocol number 11-09-221).

Reviewer #1 (Public review): https://doi.org/10.7554/eLife.99010.3.sa1
Reviewer #2 (Public review): https://doi.org/10.7554/eLife.99010.3.sa2
Author response https://doi.org/10.7554/eLife.99010.3.sa3

## Additional files

### Supplementary files
MDAR checklist

### Data availability
All data generated or analysed during this study are included in the manuscript and supporting files.

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

# Appendix 1

## Appendix 1—key resources table

| Reagent type (species) or resource | Designation | Source or reference | Identifiers | Additional information |
|---|---|---|---|---|
| Antibody | Anti-human CD14 APC/Cyanine7 (Mouse monoclonal) | BioLegend | Cat#:301820 RRID:AB_493694 | (1:50) |
| Antibody | Anti-human CD159 PE (Mouse monoclonal) | Beckman Coulter | Cat#:IM3291U | (1:25) |
| Antibody | Anti-human CD16 eFluor 450 (Mouse monoclonal) | eBioscience | Cat#:48-0168-42 RRID:AB_1272052 | (1:50) |
| Antibody | Anti-human CD183 Alexa Fluor 647 (Mouse monoclonal) | BioLegend | Cat#:353712 RRID:AB_10962946 | (1:10) |
| Antibody | Anti-human CD19 APC-eFluor 780 (Mouse monoclonal) | eBioscience | Cat#:47-0199-42 RRID:AB_1582230 | (1:25) |
| Antibody | Anti-human CD3 APC/Cyanine7 (Mouse monoclonal) | BioLegend | Cat#:300426 RRID:AB_830754 | (1:50) |
| Antibody | Anti-human CD3 eFluor 450 (Mouse monoclonal) | eBioscience | Cat#:48-0038-42 RRID:AB_1518798 | (1:50) |
| Antibody | Anti-human CD314 Brilliant Violet 421 (Mouse monoclonal) | BioLegend | Cat#:320822 RRID:AB_2566510 | (1:10) |
| Antibody | Anti-human CD335 PE (Mouse monoclonal) | BioLegend | Cat#:331908 RRID:AB_1027679 | (1:10) |
| Antibody | Anti-human CD336 PE (Mouse monoclonal) | BioLegend | Cat#:3325108 RRID:AB_756099 | (1:80) |
| Antibody | Anti-human CD337 Alexa Fluor 647 (Mouse monoclonal) | BioLegend | Cat#:325212 RRID:AB_2149448 | (1:20) |
| Antibody | Anti-human CD56 PE-Cyanine7 (Mouse monoclonal) | Invitrogen | Cat#:25-0567-42 RRID:AB_11041529 | (1:25) |
| Antibody | Anti-human HLA-DR FITC (Mouse monoclonal) | BioLegend | Cat#:307632 RRID:AB_1089142 | (1:100) |
| Antibody | Anti-human HLA-DR eFluor 450 (Mouse monoclonal) | eBioscience | Cat#:48-9952-42 RRID:AB_1603291 | (1:50) |
| Antibody | Anti-human IFN-g Alexa Fluor 647 (Mouse monoclonal) | BioLegend | Cat#:502516 RRID:AB_493031 | (1:20) |
| Antibody | Anti-human NKG2C APC (Mouse monoclonal) | R&D Systems | Cat#:FAB138A | (1:10) |
| Antibody | Anti-human Perforin PE (Mouse monoclonal) | eBioscience | Cat#:12-9994-42 RRID:AB_10854416 | (1:50) |
| Antibody | Anti-mouse/human CD45R/B220 APC/Cyanine7 (Rat monoclonal) | BioLegend | Cat#:103224 RRID:AB_313006 | (1:100) |
| Antibody | Anti-mouse CCR5 PE (Mouse monoclonal) | eBioscience | Cat#:12-1951-82 RRID:AB_657684 | (1:100) |
| Antibody | Anti-mouse CCR7 PE (Rat monoclonal) | BioLegend | Cat#:120106 RRID:AB_389357 | (1:50) |
| Antibody | Anti-mouse CCR7 Alexa Fluor 647 (Rat monoclonal) | BioLegend | Cat#:120109 RRID:AB_389235 | (1:50) |
| Antibody | Anti-mouse CD11b BUV805 (Rat monoclonal) | BD Biosciences | Cat#:741934 | (1:400) |
| Antibody | Anti-mouse/human CD11b PerCP/Cyanine5.5 (Rat monoclonal) | BioLegend | Cat#:101228 RRID:AB_893232 | (1:400) |
| Antibody | Anti-mouse CD11c PE/Cyanine5 (Armenian Hamster monoclonal) | BioLegend | Cat#:117316 RRID:AB_493566 | (1:100) |

*Appendix 1 Continued on next page*

*Appendix 1 Continued*

| Reagent type (species) or resource | Designation | Source or reference | Identifiers | Additional information |
|---|---|---|---|---|
| Antibody | Anti-mouse CD11c PE (Armenian Hamster monoclonal) | BioLegend | Cat#:117308 RRID:AB_313776 | (1:200) |
| Antibody | Anti-mouse CD11c Super Bright 600 (Armenian Hamster monoclonal) | eBioscience | Cat#:63-0114-82 RRID:AB_2722930 | (1:100) |
| Antibody | Anti-mouse CD11c Super Bright 645 (Armenian Hamster monoclonal) | eBioscience | Cat#:64-0114-82 RRID:AB_2717081 | (1:100) |
| Antibody | Anti-mouse CD19 APC/Fire 810 (Rat monoclonal) | BioLegend | Cat#:115578 RRID:AB_2892274 | (1:400) |
| Antibody | Anti-mouse CD19 FITC (Rat monoclonal) | BioLegend | Cat#:115506 RRID:AB_313640 | (1:400) |
| Antibody | Anti-mouse CD19 Alexa Fluor 700 (Rat monoclonal) | BioLegend | Cat#:115528 RRID:AB_493734 | (1:1600) |
| Antibody | Anti-mouse CD19 Brilliant Ultra Violet 615 (Rat monoclonal) | eBioscience | Cat#:366-0193-82 RRID:AB_2925404 | (1:100) |
| Antibody | Anti-mouse CD26 BUV661 (Rat monoclonal) | BD | Cat#:741492 | (1:100) |
| Antibody | Anti-mouse/rat/humanCD27 PE/Cyanine7 (Armenian Hamster monoclonal) | BioLegend | Cat#:124216 RRID:AB_10639726 | (1:100) |
| Antibody | Anti-mouse CD4 APC/Fire 810 (Rat monoclonal) | BioLegend | Cat#:100480 RRID:AB_2860583 | (1:400) |
| Antibody | Anti-mouse CD40 PE/Cyanine7 (Rat monoclonal) | BioLegend | Cat#:124622 RRID:AB_10897812 | (1:50) |
| Antibody | Anti-mouse CD44 BUV563 (Rat monoclonal) | BD Biosciences | Cat#:741227 | (1:100) |
| Antibody | Anti-mouse/human CD44 Brilliant Violet 510 (Rat monoclonal) | BioLegend | Cat#:103044 RRID:AB_2561391 | (1:80) |
| Antibody | Anti-mouse CD45 FITC (Rat monoclonal) | BioLegend | Cat#:103108 RRID:AB_312972 | 3 ug/mouse |
| Antibody | Anti-mouse CD45.2 BUV395 (mouse monoclonal) | BD Biosciences | Cat#:564616 | (1:100) |
| Antibody | Anti-mouse CD62L BV711 (Rat monoclonal) | BD Biosciences | Cat#:740660 | (1:800) |
| Antibody | Anti-mouse CD63 Alexa Fluor 647 (Rat monoclonal) | BioLegend | Cat#:143922 RRID:AB_2832513 | (1:100) |
| Antibody | Anti-mouse CD64 PE/Dazzle 594 (mouse monoclonal) | BioLegend | Cat#:139320 RRID:AB_2566558 | (1:50) |
| Antibody | Anti-mouse CD64 PE (mouse monoclonal) | BioLegend | Cat#:139304 RRID:AB_10613467 | (1:50) |
| Antibody | Anti-mouse CD80 PE-Cyanine5 (Armenian Hamster monoclonal) | eBioscience | Cat#:15-0801-82 RRID:AB_468774 | (1:100) |
| Antibody | Anti-mouse CD86 BUV737 (Rat monoclonal) | BD Biosciences | Cat#:741737 | (1:100) |
| Antibody | Anti-mouse CD86 PE/Dazzle 594 (Rat monoclonal) | BioLegend | Cat#:105042 RRID:AB_2566409 | (1:200) |
| Antibody | Anti-mouse CD8a Brilliant Violet 570 (Rat monoclonal) | BioLegend | Cat#:100740 RRID:AB_10897645 | (1:200) |
| Antibody | Anti-mouse CD8a Brilliant Ultra Violet 563 (Rat monoclonal) | eBioscience | Cat#:365-0081-82 RRID:AB_2920971 | (1:100) |
| Antibody | Anti-mouse CD8a Super Bright 645 (Rat monoclonal) | eBioscience | Cat#:64-0081-82 RRID:AB_2662353 | (1:100) |

*Appendix 1 Continued on next page*

*Appendix 1 Continued*

| Reagent type (species) or resource | Designation | Source or reference | Identifiers | Additional information |
|---|---|---|---|---|
| Antibody | Anti-mouse CXCR3 PE (Armenian Hamster monoclonal) | eBioscience | Cat#:12-1831-82 RRID:AB_1210734 | (1:100) |
| Antibody | Anti-mouse CXCR6 PE (Rat monoclonal) | BioLegend | Cat#:151103 RRID:AB_2566545 | (1:100) |
| Antibody | anti-mouse DNAM-1 PE (Rat monoclonal) | BioLegend | Cat#:128806 RRID:AB_1186119 | (1:100) |
| Antibody | Anti-EOMES PerCP-eFluor 710 (Rat monoclonal) | eBioscience | Cat#:46-4877-42 RRID:AB_2573759 | (1:50) |
| Antibody | Anti-mouse Foxp3 PE-Cyanine5 (Rat monoclonal) | eBioscience | Cat#:15-5773-82 RRID:AB_468806 | (1:100) |
| Antibody | Anti-human/mouse Granzyme B Pacific Blue (Mouse monoclonal) | BioLegend | Cat#:515408 RRID:AB_2562195 | (1:50) |
| Antibody | Anti-human/mouse Granzyme B PE (Mouse monoclonal) | BioLegend | Cat#:372208 RRID:AB_2687031 | (1:50) |
| Antibody | Anti-mouse H-2 Class I BV750 (Rat monoclonal) | BD Biosciences | Cat#:749712 | (1:400) |
| Antibody | Anti-mouse H-2D b PE (Mouse monoclonal) | BioLegend | Cat#:111508 RRID:AB_313512 | (1:50) |
| Antibody | Anti-mouse H-2Kb APC (Mouse monoclonal) | eBioscience | Cat#:17-5958-82 RRID:AB_1311280 | (1:400) |
| Antibody | Anti-mouse H-2Kb/H2Db Alexa Fluor 647 (Mouse monoclonal) | BioLegend | Cat#:114612 RRID:AB_492931 | (1:100) |
| Antibody | Anti-mouse I-A/I-E BUV496 (Rat monoclonal) | BD Biosciences | Cat#:750281 | (1:100) |
| Antibody | Anti-mouse I-A/I-E BUV615 (Rat monoclonal) | BD Biosciences | Cat#:751570 | (1:400) |
| Antibody | Anti-mouse I-A/I-E PerCP/Cyanine5.5 (Rat monoclonal) | BioLegend | Cat#:107626 RRID:AB_2191071 | (1:400) |
| Antibody | Anti-mouse IFN-g Brilliant Ultra Violet 737 (Rat monoclonal) | eBioscience | Cat#:367-7311-82 RRID:AB_2896044 | (1:50) |
| Antibody | Anti-mouse IFN-g PE (Rat monoclonal) | BioLegend | Cat#:505808 RRID:AB_315401 | (1:100) |
| Antibody | Anti-mouse IL-12/23p40 APC (Rat monoclonal) | BioLegend | Cat#:505206 RRID:AB_315369 | (1:200) |
| Antibody | Anti-mouse Ki67 FITC (Rat monoclonal) | BioLegend | Cat#:652409 RRID:AB_2562140 | (1:50) |
| Antibody | Anti-mouse Lag-3 Brilliant Violet 785 (Rat monoclonal) | BioLegend | Cat#:125219 RRID:AB_2566571 | (1:25) |
| Antibody | Anti-mouse Ly108 BUV661 (Mouse monoclonal) | BD Biosciences | Cat#:741679 RRID:AB_2871064 | (1:200) |
| Antibody | Anti-mouse Ly49A FITC (Rat monoclonal) | BioLegend | Cat#:116805 RRID:AB_313756 | (1:400) |
| Antibody | Anti-mouse Ly49D FITC (Rat monoclonal) | BioLegend | Cat#:138303 RRID:AB_10588709 | (1:1000) |
| Antibody | anti-mouse Ly49G2 FITC (Rat monoclonal) | eBioscience | Cat#:11-5781-82 RRID:AB_763604 | (1:400) |
| Antibody | Anti-mouse Ly49H APC (Mouse monoclonal) | BioLegend | Cat#:144712 RRID:AB_2783111 | (1:400) |
| Antibody | Anti-mouse Ly49I FITC (Mouse monoclonal) | eBioscience | Cat#:11-5895-82 RRID:AB_465301 | (1:100) |

*Appendix 1 Continued on next page*

*Appendix 1 Continued*

| Reagent type (species) or resource | Designation | Source or reference | Identifiers | Additional information |
|---|---|---|---|---|
| Antibody | Anti-mouse Ly6C PerCP (Rat monoclonal) | BioLegend | Cat#:128028 RRID:AB_10897805 | (1:200) |
| Antibody | Anti-mouse Ly6G Brilliant Violet 785 (Rat monoclonal) | BioLegend | Cat#:127645 RRID:AB_2566317 | (1:100) |
| Antibody | Mouse IgG1, κκPacific Blue (Mouse monoclonal) | BioLegend | Cat#:400151 RRID:AB_2923473 | Isotype control, same titer used as specific Ab |
| Antibody | Mouse IgG1, PerCP-eFluor 710 (Mouse monoclonal) | eBioscience | Cat#:46-4714-82 RRID:AB_1834453 | Isotype control, same titer used as specific Ab |
| Antibody | Mouse IgG1, κeFluor 660 (Mouse monoclonal) | eBioscience | Cat#:50-4714-82 RRID:AB_10597301 | Isotype control, same titer used as specific Ab |
| Antibody | Anti-mouse NK1.1 PE/Cyanine7 (Mouse monoclonal) | BioLegend | Cat#:108714 RRID:AB_389363 | (1:150) |
| Antibody | Anti-mouse NK1.1 PerCP/Cyanine5.5 (Mouse monoclonal) | BioLegend | Cat#:108728 RRID:AB_2132705 | (1:50) |
| Antibody | Anti-mouse NK1.1 Brilliant Ultra Violet 496 (Mouse monoclonal) | eBioscience | Cat#:364-5941-82 RRID:AB_2925353 | (1:100) |
| Antibody | Anti-mouse NKG2A PerCP-eFluor 710 (Mouse monoclonal) | eBioscience | Cat#:46-5897-82 RRID:AB_2573799 | (1:50) |
| Antibody | Anti-mouse NKG2D PE (Rat monoclonal) | eBioscience | Cat#:12-5882-82 RRID:AB_465996 | (1:50) |
| Antibody | Anti-mouse PD-1 eFluor 450 (Rat monoclonal) | eBioscience | Cat#:48-9981-82 RRID:AB_11150068 | (1:50) |
| Antibody | Anti-mouse PDCA1 Alexa Fluor 700 (Rat monoclonal) | BioLegend | Cat#:127038 RRID:AB_2819861 | (1:400) |
| Antibody | Anti-mouse PDCA1 Brilliant Violet 650 (Rat monoclonal) | BioLegend | Cat#:127019 RRID:AB_2562477 | (1:100) |
| Antibody | Anti-mouse PD-L1 Super Bright 600 (Rat monoclonal) | eBioscience | Cat#:63-5982-82 RRID:AB_2688101 | (1:200) |
| Antibody | Anti-mouse Rae-1 PE (Rat monoclonal) | R&D Systems | Cat#:FAB17582P | (1:100) |
| Antibody | Rat IgG 2 a, κ PE-Cyanine5 (Rat monoclonal) | eBioscience | Cat#:15-4321-82 RRID:AB_470140 | Isotype control, same titer used as specific Ab |
| Antibody | Rat IgG1, κ APC (Rat monoclonal) | BioLegend | Cat#:400412 RRID:AB_326518 | Isotype control, same titer used as specific Ab |
| Antibody | Rat IgG1, κ PE (Rat monoclonal) | BioLegend | Cat#:400408 RRID:AB_326514 | Isotype control, same titer used as specific Ab |
| Antibody | Rat IgG1, κBrilliant Ultra Violet 737 (Rat monoclonal) | eBioscience | Cat#:367-4301-81 RRID:AB_2896005 | Isotype control, same titer used as specific Ab |
| Antibody | Anti-mouse siglecF BV750 (Rat monoclonal) | BD Biosciences | Cat#:747316 | (1:200) |
| Antibody | Anti-mouse siglecF BV510 (Rat monoclonal) | BD Biosciences | Cat#:740158 | (1:100) |
| Antibody | Anti-mouse siglecF Super Bright 780 (Rat monoclonal) | eBioscience | Cat#:78-1702-82 RRID:AB_2744908 | (1:200) |
| Antibody | Anti-human/mouse T-bet eFluor 660 (Mouse monoclonal) | eBioscience | Cat#:50-5825-82 RRID:AB_10596655 | (1:25) |
| Antibody | Anti-mouse TCRb BUV496 (Armenian Hamster monoclonal) | BD Biosciences | Cat#:749915 | (1:50) |
| Antibody | Anti-mouse TCRb PE/Dazzle 594 (Armenian Hamster monoclonal) | BioLegend | Cat#:109240 RRID:AB_2565654 | (1:100) |

*Appendix 1 Continued on next page*

Appendix 1 Continued

| Reagent type (species) or resource | Designation | Source or reference | Identifiers | Additional information |
|---|---|---|---|---|
| Antibody | Anti-mouse TCRb-FITC (Armenian Hamster monoclonal) | homemade | | (1:200) |
| Antibody | Anti-mouse TCRd (Armenian Hamster monoclonal) | homemade | | (1:500) |
| Antibody | Anti-mouseTim-3 PE/Cyanine7 (Rat monoclonal) | BioLegend | Cat#:119716 RRID:AB_2571932 | (1:200) |
| Antibody | Anti-mouse TRAIL PE (Rat monoclonal) | BioLegend | Cat#:109306 RRID:AB_2205927 | (1:100) |
| Antibody | Anti-mouse/rat XCR1 Brilliant Violet 421 (Mouse monoclonal) | BioLegend | Cat#:148216 RRID:AB_2565230 | (1:100) |
| Antibody | Anti-mouse/rat XCR1Brilliant Violet 785 (Mouse monoclonal) | BioLegend | Cat#:148225 RRID:AB_2783119 | (1:400) |
| Sequence-based reagent | *Cxcl9*_F | This paper | qRT-PCR primer | GGAGTTCGAGGAACCCTAGTG |
| Sequence-based reagent | *Cxcl9*_R | This paper | qRT-PCR primer | GGGATTTGTAGTGGATCGTGC |
| Sequence-based reagent | *Cxcl10*_F | This paper | qRT-PCR primer | CCAAGTGCTGCCGTCATTTTC |
| Sequence-based reagent | *Cxcl10*_R | This paper | qRT-PCR primer | GGCTCGCAGGGATGATTTCAA |
| Sequence-based reagent | *Cxcl11*_F | This paper | qRT-PCR primer | GGCTTCCTTATGTTCAAACAGGG |
| Sequence-based reagent | *Cxcl11*_R | This paper | qRT-PCR primer | GCCGTTACTCGGGTAAATTACA |
| Sequence-based reagent | *Ccl3*_F | This paper | qRT-PCR primer | TTCTCTGTACCATGACACTCTGC |
| Sequence-based reagent | *Ccl3*_R | This paper | qRT-PCR primer | CGTGGAATCTTCCGGCTGTAG |
| Sequence-based reagent | *Ccl4*_F | This paper | qRT-PCR primer | TTCCTGCTGTTTCTCTTACACCT |
| Sequence-based reagent | *Ccl4*_R | This paper | qRT-PCR primer | CTGTCTGCCTCTTTTGGTCAG |
| Sequence-based reagent | *Ccl5*_F | This paper | qRT-PCR primer | GCTGCTTTGCCTACCTCTCC |
| Sequence-based reagent | *Ccl5*_R | This paper | qRT-PCR primer | TCGAGTGACAAACACGACTGC |
| Sequence-based reagent | *Cxcl16*_F | This paper | qRT-PCR primer | CCTTGTCTCTTGCGTTCTTCC |
| Sequence-based reagent | *Cxcl16*_R | This paper | qRT-PCR primer | TCCAAAGTACCCTGCGGTATC |
| Sequence-based reagent | *Cypa*_F | This paper | qRT-PCR primer | GAGCTGTTTGCAGACAAAGTTC |
| Sequence-based reagent | *Cypa*_R | This paper | qRT-PCR primer | CCCTGGCACATGAATCCTGG |

