## [Editor Report · eLife Assessment]

In this **important** study the authors develop an elegant lung metastasis mouse model that closely mimics the events in human patients. They provide **convincing** evidence for the effectiveness of IL-15/12-conditioned NK cells in this design, which was also critical for the authors being able to conclusively reveal the T cell-dependency of NK-cell-mediated long-term control of experimental metastasis. Of note, an investigator-initiated clinical trial demonstrated that similar NK cell infusions in cancer patients after resections were safe and showed signs of efficacy, which is of promising clinical application value.

---

## [Referee Report · Reviewer #1 (Public review)]

Summary:

This is a very nice paper in which the authors addressed the potential for NK cell cellular therapy to treat and potentially eliminate previously established metastases after surgical resections, which are a major cause of death in human cancer patients. To do so they developed a model using the EO771 breast cancer cell line, in which they establish and then resect tumors and the draining lymph node, after which the majority of mice eventually succumb to metastatic disease. They found that when the initiating tumors were resected when still relatively small, adoptive transfers of IL-15/12-conditioned NK cells substantially enhanced the survival of tumor-bearing animals. They then delved into the cellular mechanisms involved. Interestingly and somewhat unexpectedly, the therapeutic effect of the transferred NK cells was dependent on the host's CD8+ T cells. Accordingly, the NK cell therapy contributed to the formation of tumor-specific CD8+ T cells, which protected the recipient animals against tumor re-challenge and were effective in protecting mice from tumor formation when transferred to naive mice. Mechanistically, they used Ifng knockout NK cells to provide evidence that IFNgamma produced by the transferred NK cells was crucial for the accumulation and activation of DCs in the metastatic lung, including expression of CD86, CD40 and MHC genes. In turn, IFNgamma production by NK cells was essential for the induced accumulation of activated CD8 effector T cells and stem cell-like CD8 T cells in the metastatic lung. The authors then expanded their findings from the mouse model to a small clinical trial. They found that inoculations of IL-15/12-conditioned autologous NK cells in patients with various malignancies after resection was safe and showed signs of efficacy.

Strengths:

- Monitoring of long-term metastatic disease and survival after resection used in this paper is a physiological model that closely resembles clinical scenarios more than the animal models usually used, a great strength of the approach.

- Previous literature focused on the notion that NK cells clear metastatic lesions directly, within a short period. The authors' use of a more relevant model and time frame revealed the previously unexplored T cell-dependent mechanism of action of infused NK cells for long-term control of metastatic diseases.

- Also important, the paper provides solid evidence for the contribution of IFNgamma produced by NK cells for activation of dendritic cells and T cells. This is an interesting finding that provokes additional questions concerning the action of the interferon gamma in this context.

- The results from the clinical trial in cancer patients based on the same type of IL-15/12-conditioned NK cell infusions, was encouraging with respect to safety and showed signals of efficacy, which support the translatability of the author's findings.

Future studies in this model could shed even more light on the mechanisms. The authors do not address whether the IL-12 in their cocktail is essential for the effects they see. Relatedly, it was of interest that despite the effectiveness of the transferred IL-15/IL-12 cultured NK cells, the cells failed to persist very long after transfer. Published studies have reported that so-called memory-like NK cells, which are pre-activated with a cocktail of IL-12, IL-18 and IL-15, persist much longer in lympho-depleted mice and patients than IL-2 cultured NK cells. It would be illuminating to compare these two types of NK cell products in the author's model system, and with, or without, lymphodepletion, to identify the critical parameters. If greater persistence occurred with the memory-like NK cell product, it is possible that the NK cells might provide greater benefit, including by directly targeting the tumor.

---

## [Referee Report · Reviewer #2 (Public review)]

Summary:

The authors show convincing data that increasing NK cell function/frequency can reduce development and progression of metastatic disease after primary tumor resection.

Strengths:

The inclusion of a first-in-human trial highlighting some partial responses of metastatic patients treated with in vitro expanded NK cells is tantalising. It is difficult to perform trials in preventing further metastasis since the timelines are very protracted but more data like these highlighting a role for NK cells in improving local cDC1/T cells anti-tumor immunity will encourage deeper thinking around therapeutic approaches to target endogenous NK cells to achieve the same.

Weaknesses:

As always, more patient data would help increase confidence around the human relevance of the approach.

Comments on revisions:

The authors have addressed all my queries

---

## [Author Response]

The following is the authors’ response to the original reviews.

**Author Response**

**Reviewer #1 (Public Review):**
Weaknesses:- Having demonstrated that NK cell IFNgamma is important for recruiting and activating DCs and T cells in their model, one is left to wonder whether it is important for the therapeutic effect, which was not tested.

We conducted a preliminary study to compare the pro-survival effect of WT NK and *Ifng-/-* NK cell therapies. We found that, in the 95-500 mg day-21 tumor group, the overall survival (OS) of mice receiving *Ifng-/-* NK cell therapy significantly decreased (p = 0.045) compared to mice receiving WT NK cell therapy up to 60 days after tumor inoculation, but there was no difference in OS beyond 65 days after tumor inoculation. Therefore, we have added the following sentences at the end of the second paragraph in our Discussion (Page 32):

“However, although *Ifng*-/- NK cells induced less cDC activation compared to WT NK cells, the levels of CD86 on cDCs of mice that received *Ifng*-/- NK cells were higher than those of mice not subjected to NK cell transfer (Figure 4B). This outcome indicates the presence of IFN-g-independent or/and compensatory mechanism(s) for cDC activation by the transferred NK cells, which is in line with our preliminary result that *Ifng*-/- NK cell therapy does not significantly diminish the pro-survival effect in comparison to WT NK cell therapy beyond 60 days after tumor cell inoculation (data not shown).”

- It was somewhat difficult to gauge the clinical trial results because the trial was early stage and therefore not controlled. Evaluation of the results therefore relies on historical comparisons. To evaluate how encouraging the results are, it would be valuable for the authors to provide some context on the prognoses and likely disease progression of these patients at the time of treatment.

We had already indicated in our Results that all six patients had an ECOG performance status of 0 (Page 25 and Table). We have now added in the Results that they had “a predicted survival of >3 months” (Page 25).

**Reviewer #1 (Recommendations For The Authors):**
Minor points:(1) It would be helpful if the authors provided a rationale for why they derived their NK cell product from bone marrow cells instead of the more common source, spleen cells.

We now clarify that: “We used BM cells instead of splenocytes for NK cell culture because removal of T cells from BM cells before culturing is not necessary” (Page 35) to the section Ex vivo expansion of murine and human NK cells in our Materials and Methods.

(2) It would have been helpful to provide summary results from replicates of the cytokine production data shown in Figure 1F.

We have now added a graphical panel on the relative ΔMFI of two independent experiments to Figure 1F and revised the figure legend accordingly (Page 7—8).

(3) The role of conventional CD4+ T cells is a little unclear. The authors state in the discussion that they contribute to the antitumor response, which is consistent with their finding that depleting both CD4 T cells and CD8 T cells has a greater effect than depleting CD8 T cells. Depleting CD4 T cells alone trended towards improving the response, however. Probably Tregs are the culprit in the latter effect but a sentence or two would be helpful if the claim for a protective role for CD4 T cells is to remain.

We have now re-analyzed the data of Figure 3D by separating mice into two groups according to day 21 tumor weight, i.e., 95-600 mg and >600 mg (Page 13—14). We have revised our explanation of the Figure 3D data in the Results (Page 11—12) as follows:

“Accordingly, we examined the role of T cells in NK cell therapy by depleting T cell subsets with antiCD4 or/and anti-CD8 antibodies two days before primary tumor resection (Figure 3D Schema and Figure 3-figure supplement 1). In the 95-600 mg tumor group, depletion of CD8+ cells alone or both CD4+ and CD8+ cells diminished the effect of NK cell therapy, whereas depletion of CD4+ cells alone did not affect OS (Figure 3D). This result indicates that CD8+ T cells are essential for the effect of NK cell therapy. In contrast, the >600 mg tumor group displayed a limited NK-cell treatment effect as expected, but did exhibit improved OS upon depleting CD4+ cells alone (Figure 3D). As the proportion of lung Foxp3+CD4+ T cells in CD45+ cells positively correlated with day 21 tumor weight (data not shown), depletion of Foxp3+CD4+ T cells by anti-CD4 antibody likely has a stronger effect in augmenting the immune response for the >600 mg tumor group than the 95-600 mg tumor group. Moreover, both tumor groups showed diminished OS upon depletion of both CD4+ and CD8+ cells than was the case for depletion of CD8+ cells alone, indicating a CD8+ T cell-independent anti-tumor effect of CD4+ T cells (Figure 3D).”

(4) The schema in Figure 3E states that mice were inoculated with either EO771 tumor cells or B16F10 tumor cells, but it appears that the data only show EO771 tumor challenges. This should be corrected.

Corrected according to the reviewer’s comment.